SPECIAL ISSUE
LIFELONG DEVELOPMENT

# Planarian microtubules form a network within muscle and regulate injury-induced genes essential for regeneration patterning

Xavier N. Anderson[1] and Christian P. Petersen[1,2,*]

## ABSTRACT

Planarian muscle produces key wound signal patterning whole-body regeneration. Within muscle, generic induction of *wnt1* promotes tail regeneration, while polarized expression of the Wnt inhibitor *notum* at anterior-facing wounds drives head regeneration. Classic experiments indicate that microtubules are also involved in blastema fating, but the cell biology of planarian muscle is still poorly understood. We raised an antibody to muscle-expressed TUBA-2 and found that planarian muscle possesses a microtubule network linking contractile fibers with their mononucleated cell bodies. Microtubules were required for muscle fiber regrowth across wound sites at times that correlated with expression of wound-induced genes. Expression profiling found that sublethal colchicine treatment disrupted a subset of muscle-expressed injury-induced genes, with strongest effects on *wnt1* and *notum*. Higher colchicine doses (>200 µg/ml) prevented *wnt1* and *notum* expression, while, surprisingly, lower doses (125 µg/ml) elevated *notum* at posterior-facing wounds, thereby implicating microtubules in both the activation and polarization of genes expressed from injured muscle. Furthermore, microtubules functionally interact with Wnts to control head/tail determination. Together, planarian microtubules act in specific regulatory pathways to express key muscle-expressed and injury-induced factors used for blastema fating.

KEY WORDS: Planaria, Regeneration, Microtubule, Wnt, Muscle

## INTRODUCTION

Animals capable of whole-body regeneration must determine the identity of lost tissues in order to execute appropriate growth programs. Planarians accomplish whole-body regeneration through control of pluripotent neoblast stem cells, which can differentiate into all adult cell types, conserved transcriptional injury responses and robust positional information (Reddien, 2018). Standing Wnt and BMP gradients are expressed regionally within muscle and determine the regional identities necessary to regenerate (Molina et al., 2007; Reddien et al., 2007; Petersen and Reddien, 2008; Gurley et al., 2010; Witchley et al., 2013; Hill and Petersen, 2015; Lander and Petersen, 2016; Scimone et al., 2016, 2018; Sureda-Gomez et al., 2016;

Stuckemann et al., 2017; Clark and Petersen, 2023). Muscle is also injury responsive and expresses factors regulating blastema identity (Scimone et al., 2017). Planarian muscle cells are mononucleated and possess actinomyosin-rich contractile fibers projecting either circularly, diagonally or longitudinally within the body wall, or dorsoventrally and surrounding the gut (Cebria et al., 1997; Cebria, 2016). However, the detailed subcellular composition of planarian muscle and how this contributes to their injury responsiveness and patterning abilities remains unknown. Planarians undergo a head-versus-tail decision early in regeneration, which is a model for understanding mechanisms of blastema fating. In *Schmidtea mediterranea*, injury triggers muscle expression of *wnt1* at any wound and expression of the Wnt inhibitor *notum* preferentially at anterior-facing wound sites (Petersen and Reddien, 2009, 2011). *wnt1* acts through *beta-catenin-1* to suppress head regeneration and promote tail formation (Gurley et al., 2008; Iglesias et al., 2008; Petersen and Reddien, 2008, 2009; Adell et al., 2009; Rink et al., 2009), while *notum* acts oppositely to promote head regeneration (Petersen and Reddien, 2011). Wound-induced *wnt1* and *notum* are expressed from 6 to 24 h in muscle near the injury, followed by expression at posterior and anterior blastema poles, respectively, from 48 to 96 h. The late expression phase requires stem cells and specific differentiation programs generating anterior and posterior blastema signaling centers (Petersen and Reddien, 2009; Gurley et al., 2010; Scimone et al., 2014; Vasquez-Doorman and Petersen, 2014; Vogg et al., 2014; Akheralie et al., 2023), while the early expression phase is stem-cell independent, takes place in pre-existing muscle and has different functional requirements. Early-phase *wnt1* activates generically in muscle at the injury site, depends on *foxG* (Pascual-Carreras et al., 2023), *arx-3* (Akheralie et al., 2023) and Hedgehog signaling (Rink et al., 2009; Yazawa et al., 2009), but not Wnt/β-catenin signaling (Petersen and Reddien, 2009), and is negatively regulated by *follistatin* (Tewari et al., 2018) and *ddx24* (Sarkar et al., 2022). Early injury-induced *notum* is expressed from longitudinal muscle (Scimone et al., 2017), and is positively regulated by *beta-catenin-1*, *wnt1*, *wntP-2* and *arx-3*, and negatively regulated by Dishevelled, *wnt11-1/-2* and *activin-2* (Petersen and Reddien, 2011; Cloutier et al., 2021; Gittin and Petersen, 2022; Akheralie et al., 2023). Therefore, *notum* is a Wnt feedback inhibitor at anterior-facing wounds, while *wnt1* is prevented from activating *notum* expression at posterior-facing wounds through a polarity mechanism contributing to the blastema head-versus-tail decision process. Inhibition of *notum* or *wnt1* only after amputation caused polarity reversal phenotypes (Petersen and Reddien, 2009, 2011), so their wound-induced expression is essential for regeneration, but it is still unresolved what relative contributions are made by their early versus late expression programs.

Muscle is pivotal for planarian regeneration, but it is unknown how muscle structure or physiology supports activation programs for factors like *wnt1* and *notum*. The length of fiber contributed by

[1]Department of Molecular Biosciences, Northwestern University, Evanston, IL 60208. [2]Robert Lurie Comprehensive Cancer Center, Northwestern University, Evanston, IL 60208.

*Author for correspondence (christian-p-petersen@northwestern.edu)

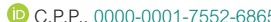 C.P.P., 0000-0001-7552-6865

each muscle cell has been difficult to assess by whole-mount immunostaining, but measurements of dissociated cells suggest projections are ~200 µm, while animals range from 1 to 20 mm (Cebria, 2016). Early activation of *wnt1* and *notum* occurs in muscle cell bodies within a similar 200 µm distance from the wound site, suggesting injured fibers could directly control this process. Alternatively, nearby tissues such as epidermis could indirectly signal to muscle, e.g. *equinox* is injury-induced in epidermis and promotes *wnt1*, *notum* and *follistatin* expression (Scimone et al., 2022). Muscle could also communicate signals distantly, and is required for transducing a body-wide wave of phospho-Erk (Fan et al., 2023). However, the lack of reagents to detect core processes in planarian muscle has limited an understanding of how muscle structure relates to its pro-regenerative function.

Classic studies of *Dugesia dorotocephala* found low doses of the microtubule depolymerizer colchicine resulted in regeneration patterning defects, including cyclopia, forked blastemas and posterior-facing heads, while high doses were lethal (McWhinnie, 1955). However, these observations have not been linked to specific pathways now known to control blastema fating. Microtubules are intrinsically polarized and have numerous functions in transport, force generation, cytoskeletal support, tissue migration and cell proliferation (Etienne-Manneville, 2013; Barlan and Gelfand, 2017; Brouhard and Rice, 2018; Goodson and Jonasson, 2018; Akhmanova and Kapitein, 2022), so they could contribute to regeneration in many possible ways. Although a major physiological consequence of microtubule inhibition across organisms is mitotic arrest (Pernice, 1889; Dixon and Malden, 1908), including in planarians (van Wolfswinkel et al., 2014), early *wnt1* and *notum* activation is independent of cell proliferation (Petersen and Reddien, 2009; Vasquez-Doorman and Petersen, 2014). Planarian microtubules are involved in ciliogenesis in epidermal and excretory cells, and have roles in the nervous system, germline and neoblasts (Thi-Kim Vu et al., 2015; Basquin et al., 2019; Magley and Rouhana, 2019; Vu et al., 2019; Lesko and Rouhana, 2020; Christman et al., 2021; Ge et al., 2021; Gambino et al., 2022; Rouhana et al., 2022). However, the relationships between microtubules and specific responses in planarian muscle are unknown. Previous efforts to detect planarian microtubules using existing reagents detect staining in other tissues, such as ciliated epidermal and excretory cells and neurons (Sánchez Alvarado and Newmark, 1999; Robb and Sánchez Alvarado, 2002; Rink et al., 2011), and the presence and/or organization of microtubules in planarian muscle is unknown. Therefore, it is unclear whether the patterning effects of colchicine originally described in planarians affect muscle.

We generated an antibody recognizing an alpha-tubulin protein expressed in planarian muscle and used it to identify a network of microtubules running in parallel to the contractile fiber that can connect to muscle cell bodies. This reagent enabled detection of muscle fiber growth early in the process of wound repair. Using microtubule depolymerizing drugs, we determine that *wnt1* and *notum* are among a small number of muscle-expressed, wound-induced genes strongly affected by microtubule disruption. Furthermore, we integrate these findings into known pathways for blastema identity determination to reveal how microtubules participate in regeneration specificity.

## RESULTS
### Planarian TUBA-2 protein marks filaments along muscle fibers
The planarian genome encodes 8 β-tubulins and 62 α-tubulins (Table S1). To examine the distribution and potential functions of microtubules in planarian muscle, we identified an α-tubulin

with highly enriched expression across planarian muscle cell types (Fig. S1A), *tuba-2* (*dd508*). We raised a rabbit polyclonal antibody predicted to uniquely target TUBA-2 protein (Fig. S1B) and found it labeled a dense filament network associated with muscle fibers (Fig. 1A). RNAi to target *tuba-2* depleted anti-TUBA-2 staining, indicating appropriate targeting (Fig. 1C). However, we did not detect any defects in regeneration or homeostasis after *tuba-2* RNAi (10/10 animals), suggesting potentially redundant functions for this protein, consistent with the large number of planarian alpha-tubulins. Co-labeling with planarian 6G10 antibody (Ross et al., 2015), which labels actinomycin-rich muscle fibers (Cebria et al., 1997), revealed that multiple TUBA-2$^+$ filaments ran in parallel to each 6G10$^+$ fiber (Fig. 1B). Higher-resolution imaging of individual z-slices detected TUBA-2 staining in circular, diagonal and longitudinal body-wall muscle fibers distributed across the body axis (Fig. 1C, Movies 1 and 2). Carnoy's fixative enabled detection of TUBA-2 protein, but 4% formaldehyde fixation did not. Carnoy's fixation also resulted in suboptimal Hoechst staining of nuclei, but we were able to identify z-slices that revealed the presence of TUBA-2 protein surrounding rare nuclei that we suggest represents the muscle cell body (Fig. 1D). Using a recently developed fixation (nitric/ formic acids and paraformaldehyde, NAFA) (Guerrero-Hernandez et al., 2024), we detected *collagen*$^+$ muscle cell bodies by FISH followed by anti-TUBA-2 immunostaining to find close association between muscle microtubules within fibers and associated cell bodies (Fig. 1E, Movies 3 and 4). Together, TUBA-2 protein is strongly expressed throughout planarian body-wall muscle.

To determine whether the filaments labeled by anti-TUBA-2 antibody are microtubules, we treated animals with chemical microtubule depolymerizers. Based on historical studies finding a role for microtubules in blastema patterning (McWhinnie, 1955), we chose a dose schedule in which animals were treated with colchicine or nocodazole for 24 h prior to fixation and immunostaining followed by recovery in drug-free media (Fig. 1F). By staining animals for the mitotic marker H3P over a dosage series of colchicine treatment, we determined that mitotic arrest reached maximal levels at 400 µg/ml (Fig. S2). In animals stained using anti-TUBA-2, doses that caused sub-maximal mitotic arrest (125 and 200 µg/ml colchicine) were sufficient to disrupt muscle fiber staining and cause accumulation of signal around cell bodies located nearby (Fig. 1F). Nocodazole administered at concentrations previously used to arrest mitosis maximally (400 ng/ml) (van Wolfswinkel et al., 2014) and also a lower dose (100 ng/ml) resulted in similar disruptions to TUBA-2 staining that reduced filament density and accumulated signal in cell bodies, and under these conditions without loss of 6G10$^+$ fibers (Fig. 1F,G). The disruptions to TUBA-2 from colchicine or nocodazole did not appear regionalized across the body axis (20/20 animals). Nocodazole caused stronger effects on TUBA-2 staining than colchicine but also greater lethality (8/95 survived 24 h of 100 ng/ml nocodazole but 28/30 survived 125 µg/ml colchicine), so we used colchicine for the majority of subsequent experiments. Together, these results indicate TUBA-2 marks muscle microtubules, and that the muscle cell body could be a site of unpolymerized tubulin heterodimer accumulation. The sensitivity of muscle microtubules to inhibitors of microtubule polymerization indicates they are in a state of dynamic maintenance.

### Microtubules are necessary for muscle fiber repair after injury
We examined how the muscle microtubules change dynamically after injury (Fig. 2A,B). To enable the most straightforward imaging, we focused on flank incisions, which heal rapidly within 1 day. In the first 2 h following incision injury, anti-TUBA-2

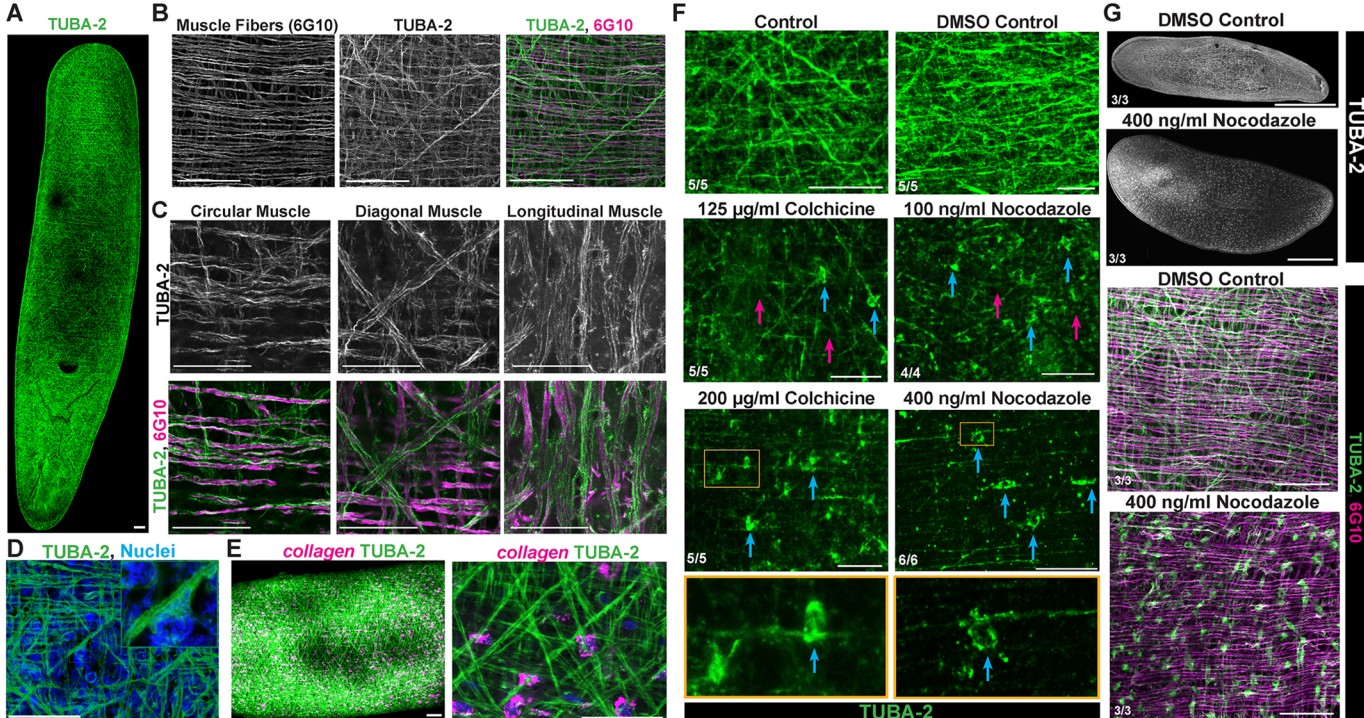

**Fig. 1. Planarian TUBA-2 marks muscle microtubules.** (A-C) Planarians were Carnoy's fixed and stained using anti-TUBA-2 antibody and/or 6G10 muscle antibody using (A) 10× (max projection), (B) 20× (max projection), (C) 40× (z-slices) objectives, showing detection on filaments running along each 6G10⁺ fiber. 6G10 preferentially stains circular and diagonal fibers; longitudinal fiber images show an additionally brightened 6G10 signal for visualization. (D) Carnoy's fixative enabled only weak detection of nuclei by Hoechst staining, but in rare cases candidate cell bodies of TUBA-2-stained muscle fibers could be identified. (E) Immunostaining TUBA-2 and FISH of *collagen*⁺ muscle cell bodies (NAFA fix). Image on the right is a magnified view of the image on the left. (F,G) Colchicine or nocodazole caused a reduction to TUBA-2 muscle fiber staining (magenta arrows) and signal accumulation in hypothesized muscle cell bodies (blue arrows). Samples were Carnoy's fixed and imaged at 20× (except 400 ng/ml nocodazole, imaged at 40×). (G) Bottom panels are higher-magnification views of the top panels. Scorings indicate the number of animals similar to representative images. Detection of TUBA-2 used fluorescent secondary antibodies (A-D,G) or tyramide amplification (E,F). Scale bars: 300 µm (G, top panels); 50 µm (A,B,D-F, bottom panels in G); 25 µm (C).

antibody detected strong signal at the wound site, but the staining had a disorganized and blebbed appearance compared to intact tissue. By 4 h, TUBA-2⁺ muscle fibers projected outward from each side of the wound, and by 6 h some fibers had crossed the injury site. By 18-24 h, the muscle fiber network had fully traversed the wound, but TUBA-2 staining remained stronger at this location (Fig. 2A). These overall behaviors were mimicked by 6G10 staining (Fig. 2B), suggesting that the growth of microtubules is likely concurrent with growth of muscle fibers across the wound. Colchicine treatment for 24 h prior to injury (at 125 µg/ml), and followed by recovery in control media after surgery, prevented any formation of TUBA-2⁺ muscle fibers projecting across the wound by 4-6 h (Fig. 2A). By 18 h after injury, wound sites from colchicine-treated animals appeared closed, but they tended to accumulate muscle cell body expression of TUBA-2 (Fig. 2A). 6G10 expression at the wound site was also disrupted by colchicine treatment (4-24 h, Fig. 2B), suggesting microtubules are likely important for the process of muscle fiber growth. We suggest that, under these conditions, the overall process of wound healing is likely be driven by other tissues, such as the overlying epidermis, but that microtubule-dependent muscle fiber growth contributes to restoration of the muscle system at wounds. Given the rapid responses to regrow muscle fibers across incisions, we predicted this process would likely be driven by growth of pre-existing muscle rather than new differentiation. To examine this possibility, we tested muscle fiber growth after incisions in animals lethally irradiated with 6000 Rads of X-rays, a treatment known to eliminate planarian stem cells and any new

differentiation (Reddien et al., 2005). Irradiated animals could restore TUBA-2⁺ muscle across an incision site with the same kinetics as uninjured animals, indicating repair likely occurs through growth of pre-existing muscle (Fig. 2C). We also stained regenerating trunk fragments with the anti-TUBA-2 antibody during the first 18 h after amputation (Fig. S3), although it was difficult to unambiguously assign TUBA-2⁺ fibers at the injury site as either body-wall muscle projecting across the amputation or DV muscle fibers parallel to the wound plane. However, we did not notice substantial differences between TUBA-2 at anterior or posterior wounds. We conclude microtubules have a likely involvement in muscle fiber growth early, within the first day following incision injuries, and detection of muscle microtubules can track the behavior of muscle during injury repair and regeneration.

## Microtubule inhibition impairs expression of a subset of injury-induced genes

Given the involvement of muscle microtubules in early injury responses, we hypothesized that microtubules might contribute to transcriptional responses to injury and used RNAseq to analyze microtubule-dependent transcriptome changes after amputation. Animals were treated with colchicine for 24 h with 0 or 200 µg/ml colchicine, then amputated and allowed to recover in colchicine-free media, followed by isolation of total RNA from wound-proximal tissue at 0, 4 and 18 h after amputation. 118 genes were identified as wound-induced in normal animals because they were upregulated at

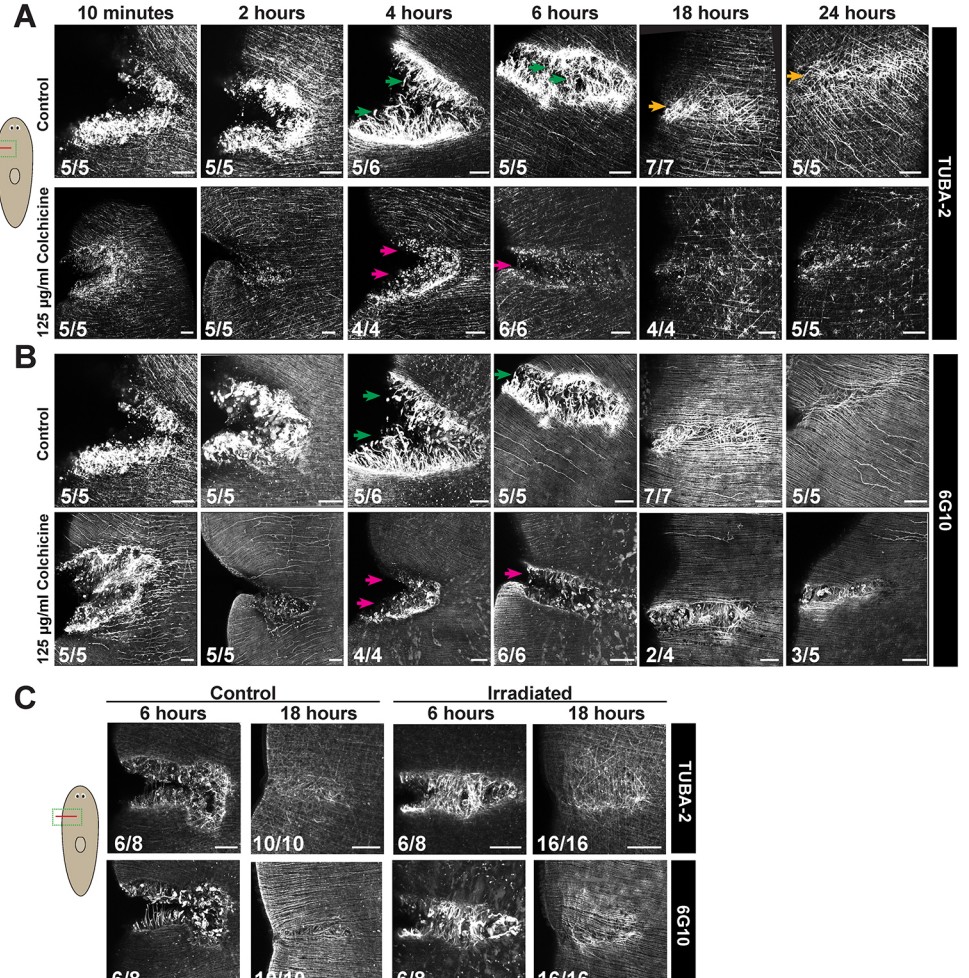

**Fig. 2. Muscle microtubules undergo dynamic changes during wound repair.** (A,B) Co-immunostaining with (A) anti-TUBA-2 and (B) 6G10 after incisions in animals treated for 24 h with 125 µg/ml colchicine and recovery in drug-free media. In controls, TUBA-2⁺ projections extended from wound sites by 4 h (green arrows), wounds appeared closed by 6 h and elevated TUBA-2 levels on microtubules crossing the wound site were detected at 18-24 h (orange arrows). 125 µg/ml colchicine treatment resulted in reduced TUBA-2 staining at wound sites, a lack of TUBA-2⁺ fibers protruding from the wound edges by 4 h (magenta arrows) and a reduction in fibers along with cell body expression by later timepoints (18-24 h). Animals treated with 125 µg/ml colchicine all healed their wounds by ~6 h. Muscle fibers protruding from wound edge at 4 h and crossing the wound site at 18 h were 6G10⁺. Colchicine-treated animals lacked 6G10⁺ fiber protrusions extending across the wound site at these times (magenta arrows), although staining was present at reduced levels away from the wound site. (C) Animals were exposed to 6000 Rads of X-rays 3 days before incision and immunostaining for anti-TUBA-2/6G10, and compared to controls. All animals succeeded in healing the TUBA-2/6G10 fiber network across the wound. Scorings show the number of animals similar to representative images. TUBA-2 was detected by tyramide amplification and 6G10 by fluorescent secondary antibodies. Scale bars: 50 µm.

either 4 or 18 h after amputation under control treatments. A large number of genes were perturbed by colchicine at the 0-h timepoint (2691 downregulated and 2348 upregulated), reflecting broad roles for microtubules in the animal (Fig. S4A, Table S2). However, a more limited number of genes were differentially expressed by colchicine treatment in regeneration (375 genes were downregulated at either 4 h or 18 h, and 278 genes were upregulated at either 4 or 18 h), likely due to allowing time for recovery in colchicine-free media following amputation (Fig. S4A, Table S2). Genes downregulated by colchicine at 0 h included *tuba-2* and many other tubulins, consistent with a conserved feedback mechanism that downregulates tubulin mRNA and translation when tubulin monomers are in excess (Cleveland et al., 1981; Batiuk et al., 2024) (Fig. S4C). Using this dataset, we assessed the behavior of wound-induced genes (Wenemoser et al., 2012; Wurtzel et al., 2015) (Fig. 3A,B, Fig. S5). Only a subset of injury-induced genes had significantly different expression at 4 or 18 h due to colchicine treatment. These included reduced expression for *wnt1*, *notum*, *runt-1* and *h2b*, while *dd1039*, *dd9204*, *HSP20L*, *TNFAF1* and *hadrian* had increased expression (Fig. 3A,B, Fig. S5). *wnt1* and *notum* are injury induced exclusively in muscle, while *runt-1* and *h2b* are induced in neoblasts (Wurtzel et al., 2015). Colchicine treatment still enabled the expression of many other injury-induced genes known to be expressed from muscle, including *inhibin-1*, *wntless*, *follistatin*, *nlg-1*, *CALM1*, *35exo* and *fascin*. We confirmed these effects on *runt-1*, *inhibin-1*, *wntless*, *nlg-1* and *follistatin* by

FISH (Fig. 3C, Fig. S6A,B). Therefore, muscle injury-induced genes displayed a variation in their sensitivity to colchicine treatment. Likewise, colchicine inhibited some but not all injury-induced genes expressed from neoblasts and epidermis. Among the genes wound-induced in neoblasts, *runt-1* and *h2b* were sensitive to colchicine but not *inx-13* and *HSP20L*, and colchicine-treated animals still activated the epidermally expressed genes *equinox* and *TNFAF*. Together, although colchicine caused a complex transcriptional response, its effects on injury-induced genes were surprisingly specific and prominently affected *wnt1* and *notum*.

We further evaluated whether the effect on *wnt1* and *notum* could be due to wound healing failure or loss of muscle. Wounds close to stop the outflow of debris within ~1-2 h and acquire a smoother appearance, and treatment with 125 and 200 µg/ml colchicine still enabled this process, as monitored in live animals (Fig. S7), consistent with ultrastructural studies finding planarian epidermal wound closure is impaired by inhibition of actin (Pascolini et al., 1984) and not microtubules (Hori, 1978). Therefore, the sensitivity of some wound-induced factors like *notum* to colchicine is unlikely to arise from a failure of wound closure. Likewise, a trivial explanation for the elimination of *notum* or *wnt1* expression could be failure to produce or sustain muscle cells capable of expressing these genes at the wound site. However, animals treated with colchicine still had abundant *collagen*⁺ body-wall muscle, including *myoD*⁺ longitudinal muscle cells, which are the sole source of *notum* (Fig. S8). We conclude that microtubules have a

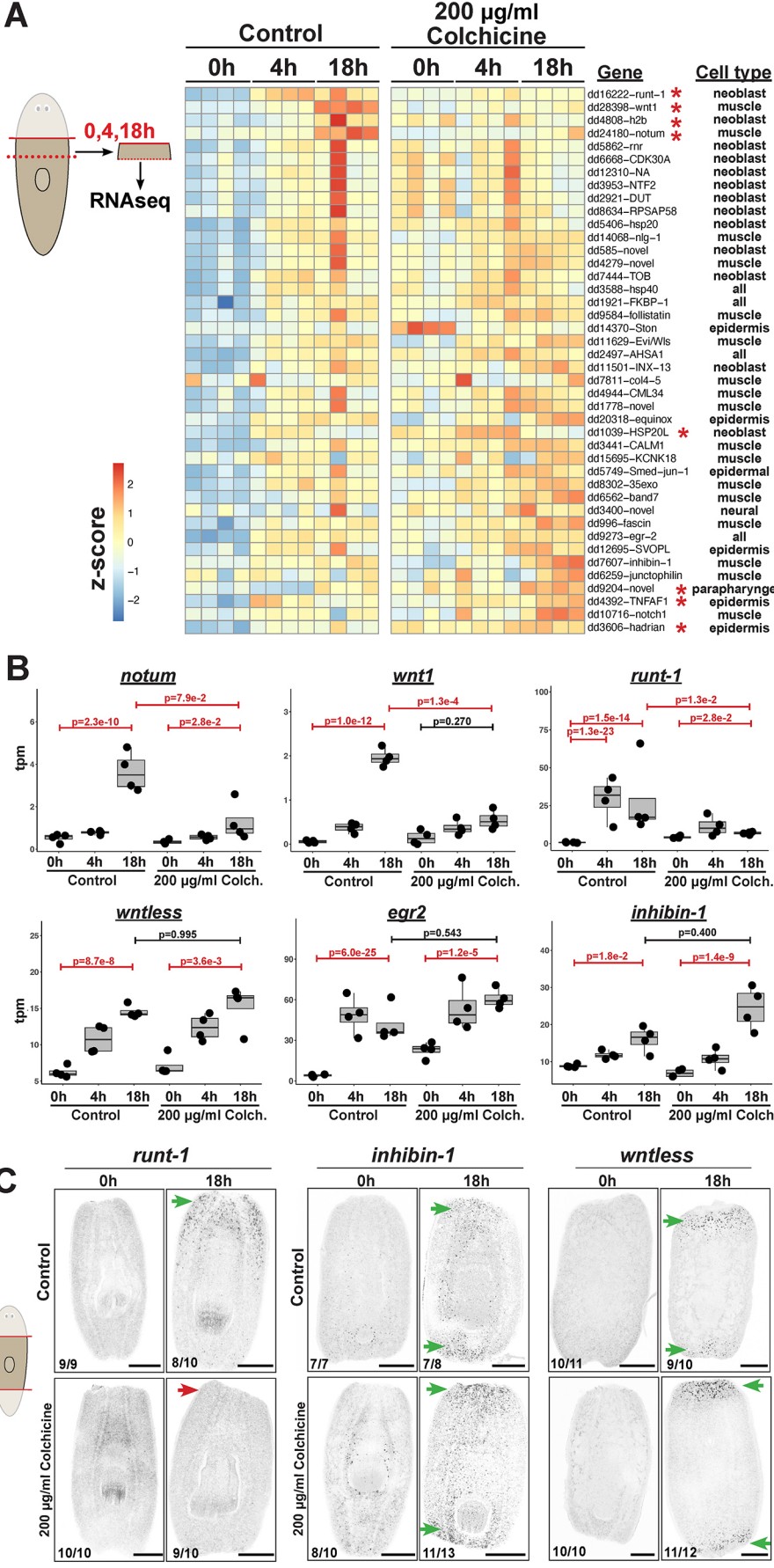

**Fig. 3. RNAseq identifies a small number of wound-induced genes strongly affected by 200 µg/ml colchicine treatment.** (A) Animals were treated with control media or media containing 200 µg/ml colchicine for 24 h, followed by head removal (red line) and recovery in drug-free media. Total RNA was then collected near anterior-facing wounds (dotted blue line) at 0, 4 and 18 h post-amputation, and analyzed by RNAseq. A heatmap displaying expression of known injury-induced genes [4 biological replicates, z-score row-scaled expression of log2(FPKM+1)]. Gene names incorporate Dresden transcriptome numbers. Red asterisks indicate a significant change at 4 h or 18 h between colchicine versus control treatments (*P*.adj<0.1). (B) Expression of individual genes (transcripts per million reads). Boxplots show medians and IQR. Values above brackets show significant (*P*.adj<0.1, red) or insignificant (*P*.adj>0.1, black) comparisons. Colchicine reduced expression of injury-induced *notum*, *wnt1* and *runt-1*, but not of *inhibin-1*, *wntless* or *egr2*. (C) FISH showing 200 µg/ml colchicine prevented injury-induced *runt-1* expression (red arrow) but enabled *inhibin-1* or *wntless* (green arrows). Scorings indicate the number of animals similar to representative images. Scale bars: 200 µm.

specific role in the capacity for muscle cells to express a subset of injury-induced genes.

## Microtubules promote expression of injury-induced *notum* and control its polarization

Given the relatively specific effect of microtubule inhibition on *notum* and *wnt1*, we carried out experiments to better understand how their expression changes spatially in response to various doses of colchicine. We began with *notum* as it is the only gene expressed asymmetrically at anterior versus posterior wound sites (Wurtzel et al., 2015). A colchicine concentration series testing *notum* expression at 18 h confirmed the RNAseq data that doses of at least 200 µg/ml led to strong reductions or failures of *notum* expression from anterior-facing wounds (Fig. S9A). Surprisingly, however, 125 µg/ml colchicine caused elevation of *notum* expression at posterior-facing wounds at ~20-40% penetrance (Fig. S9A). In separate cohorts measured across 4 independent experiments, we confirmed that 125 µg/ml colchicine treatment caused 25/59 animals to have >10 *notum*+ cells at posterior-facing wounds, while all control animals had fewer than 10 *notum*+ cells (36/36 animals) (Fig. 4A). This excess expression phenotype at posterior-facing wounds occurred between 75-125 µg/ml colchicine, but no concentration tested could fully separate from doses leading to less *notum* at anterior-facing wounds (Fig. S9A). We also verified nocodazole-treated animals also displayed similar phenotypes, with a lower dose (100 ng/ml) elevating *notum* at posterior-facing wounds, while higher doses (400 ng/ml) caused loss of *notum* from anterior-facing wounds (Fig. S9B). Therefore, microtubules likely participate in early wound-induced *notum* expression in two different ways: one that promotes *notum* expression and is only inhibited by relatively higher amounts of colchicine, and another that polarizes *notum* expression and is inhibited by relatively lower amounts of colchicine.

We examined the impact of colchicine on the pole-specific phase of *notum* expression from 48-96 h and head/tail regeneration outcomes. At both the 125 and 200 µg/ml doses, colchicine delayed expression of a *notum*+ anterior pole that began at 24 h in control animals but was not detected in any colchicine-treated fragments until 96 h (Fig. 4B). Likewise, while transverse fragments treated with 0 or 50 µg/ml colchicine produced anterior pole *sfrp-1* by 5 days, 125 µg/ml colchicine treatment delayed *sfrp-1*+ anterior pole formation until between days 5 and 16 (Fig. S10). This delay could be due to a combination of modified injury signaling and also recovery from mitotic arrest, which through H3P staining described above was more severe in the anterior half of the body (Fig. S2). Next, given the classic studies of microtubule inhibition on planarian regeneration (McWhinnie, 1955), we analyzed a series of experiments to determine whether colchicine could cause posterior head regeneration. Although posterior blastemas underwent a reproducible but incompletely penetrant elevation of *notum* expression during the early polarity decision after microtubule inhibition, we did not detect regeneration of posterior heads from animals treated over a range of colchicine concentrations (0 to 200 µg/ml) (total of *n*=80 animals as assessed by live scoring). Likewise, posterior blastemas of colchicine-treated animals did not form *notum*+ anterior pole tissue within 96 h of regeneration (Fig. 4B), and no colchicine-treated animals formed *sfrp1*+ anterior pole tissue within their tail blastemas at either 5 or 16 days post-amputation (61/61 animals stained, Fig. S10). We attempted to examine head/tail regeneration in animals treated with higher doses (200 or 500 µg/ml colchicine 24 h prior to amputation) or after dosing for longer periods of time (72-120 h of 125 or 200 µg/ml colchicine prior to amputation), but these caused lethality within

4-5 days (50 animals total). We conclude that, despite modifying the early expression of injury-induced head/tail regulatory factors, microtubule inhibition alone was not able to lead to blastema head/tail fate transformations. *notum* expression polarity correlates with head/tail regeneration outcomes in normal animals, but earlier work found that disruptions of *notum* polarity do not always correlate with head/tail regeneration outcomes. For example, a greater fraction of *activin-2(RNAi)* animals express excess *notum* than go on to form posterior-facing heads (Cloutier et al., 2021). A possible explanation is that early *notum* polarity acts redundantly with other factors to regulate blastema fating.

However, *notum* expression represents the earliest known symmetry-breaking step in early planarian regeneration, and so we sought to place microtubule regulation within this pathway. *wnt1* and *beta-catenin-1* promote *notum* expression from anterior-facing wounds (Petersen and Reddien, 2011; Gittin and Petersen, 2022), so we tested whether these factors were required for ectopic *notum* produced at posterior-facing wounds under the 125 µg/ml colchicine (Fig. 4C). Indeed, inhibition of *beta-catenin-1* or *wnt1* prevented expression of *notum* at posterior-facing wounds in animals treated with 125 µg/ml colchicine (Fig. 4C). By contrast, inhibition of *wnt11-2* increases *notum* expression from posterior-facing wounds at 18 h (Gittin and Petersen, 2022), and 125 µg/ml colchicine did not enhance this effect (Fig. S11). One possibility is that *wnt11-2* acts through a microtubule-dependent step to control *notum* polarization, consistent with the failure to observe phenotypic enhancement in this experiment. However, because both the RNAi and also microtubule inhibitions are unlikely to eliminate function, these factors could act in parallel but in some way preventing the detection of enhancement. We conclude that, at a minimum, *wnt1* and *beta-catenin-1* are important for the overactivation of *notum* at posterior-facing wounds after 125 µg/ml colchicine.

## Microtubules both promote expression of injury-induced *wnt1* and restrict midline *wnt1* to the posterior

We next examined how colchicine affected *wnt1* expression behavior (Fig. 5A). In normal animals, *wnt1* wound-induced expression normally peaks at 12 h and is declining by 18 h, followed by expression selective to the posterior pole starting at 48-72 h. By contrast, 125 or 200 µg/ml colchicine strongly reduced or eliminated expression of *wnt1* at 12 or 18 h in anterior-facing wounds (Fig. 5A). Posterior-facing wounds had a more complex response, with 200 µg/ml colchicine preventing wound-induced expression at both timepoints, while 125 µg/ml colchicine delayed the onset of *wnt1* until 18 h. In addition, posterior wound sites had expression of *wnt1* along the dorsal posterior midline, reminiscent of its expression in uninjured animals (Fig. 5A). Similarly, *wnt1* expression remained expressed at the posterior pole of colchicine-treated animals from 48 to 96 h and occupied a larger domain within the posterior domain (Fig. 5B). Homeostatic animals treated with 125 or 200 µg/ml colchicine for 3 days also underwent an expansion of the *wnt1* posterior domain (Fig. 5C). However, domain sizes of other Wnt-related factors marking the AP axis (*wnt11-2*, *wntP-2* and *notum*) were not significantly different (Fig. S12A). Further demonstrating that specificity of microtubule inhibition effects on *wnt1*, midline expression of *slit* expression was normal at 18 h in colchicine-treated animals (Fig. S12B). Because animals all died within 4-5 days of continuous colchicine treatment, it is possible that any effects of microtubule inhibition on AP patterning downstream of *wnt1* could occur over a longer timescale that cannot be observed because of animal death. However, these results indicate that microtubules can limit the posterior domain of

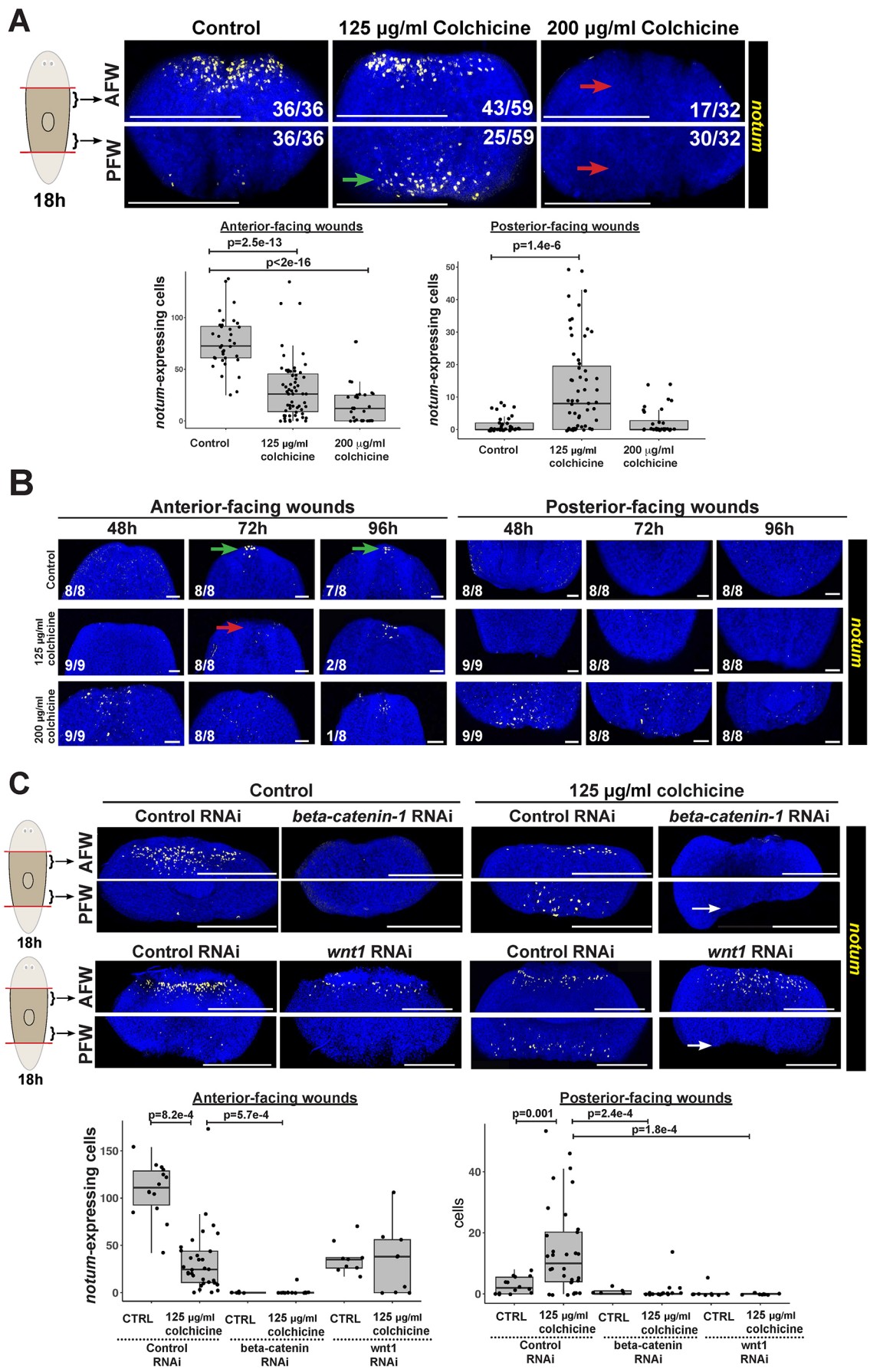

**Fig. 4.** See next page for legend.

**Fig. 4. Microtubules are required for *notum* activation and regulate *notum* polarity.** (A,B) FISH detecting *notum* following 24 h colchicine pretreatment, amputation, recovery in control media and fixation at indicated times. Plots quantify *notum*+ cells at anterior- (AFW) or posterior-facing wounds (PFW). (A) 200 µg/ml colchicine eliminated *notum* expression (red arrows), while 125 µg/ml caused upregulation of *notum* from posterior-facing wounds (green arrow). *P*-values were computed using a Kruskal–Wallis test followed by Dunnett's test comparing multiple samples against a common control. (B) Colchicine delayed the formation of a *notum*+ anterior pole until at least 96 h versus 48-72 h in control animals, and did not cause formation of a *notum*+ posterior pole at any dose/timepoint (green arrows, normal pole; red arrow, lack of pole). (C) Top: *beta-catenin-1* RNAi (6 feedings, 2 weeks) or *wnt1* RNAi (15 feedings, 4 weeks), eliminated excess *notum* expression at posterior-facing wounds caused by 125 µg/ml colchicine administered as in A (arrows). Bottom: quantifications of *notum*+ cells combined from the two experiments and *P*-values calculated using a Kruskal–Wallis test followed by Dunn's test to compare multiple samples across conditions. (A,C) Plots show numbers of *notum*+ or *wnt1*+ cells from individual animals (dots), with overlaid boxplots showing median and 25-75% interquartile range. Scorings in A-C show number of animals similar to representative images. Scale bars: 300 µm (A,C); 50 µm (B).

pre-existing and tail blastema-specific *wnt1* at the posterior pole. Together, these results indicate a complex relationship between microtubule function and both *notum* and *wnt1* expression in planarians. High colchicine doses (200 µg/ml) prevented wound-induced *wnt1*, lower doses (125 µg/ml) delayed the onset of injury-induced *wnt1* at posterior-facing wounds, and both doses still enabled regeneration of a *wnt1*+ posterior pole after amputation whose domain was expanded.

Given the role of *wnt1* in promoting *notum* expression, the dose-dependent expression behavior of *wnt1* after microtubule inhibition offers potential explanations for why lower colchicine doses result in ectopic *notum* expression but high doses lead to no *notum* expression. We suggest a single microtubule-dependent process may control the extent and/or timing of *wnt1* activation, and this process may either be under A-P control or differentially sensitive to microtubule inhibitions at distinct axis positions. High colchicine doses strongly reduce this response, leading to lack of injury-induced *wnt1* and consequently lack of *notum* expression. Lower colchicine doses still enable delayed *wnt1* expression that, at posterior-facing wounds, can activate *notum* and reveal a simultaneous disruption to the polarity ordinarily restricting *notum* to the anterior. A potentially separate microtubule-dependent process additionally regulates *wnt1* at the posterior pole. The net effect of modifying *wnt1* and *notum* after microtubule inhibition still enables appropriate tail blastema specification to occur, because the dual reductions to injury-induced *wnt1*, and/or increases in *notum* expression at posterior-facing wounds, could be counterbalanced by posterior-pole *wnt1* expression. Consistent with a model in which early injury-induced and posterior-pole *wnt1* expression phases act in parallel, a recent study found that *S. polychroa* embryos transit through a stage where they can regenerate tails and the *wnt1* posterior pole without the earlier wound-induced expression phase of *wnt1* (Booth et al., 2025).

One possible contributor to the effects observed on expression of these genes could be the particular strategy of microtubule inhibition used in these experiments. In the experiments above, we used a scheme for microtubule inhibition for 24 h prior to amputation, followed by recovery in microtubule-free media, in order to follow the design of classic studies that found microtubule inhibition could cause head/tail regeneration polarity defects (McWhinnie, 1955) and avoid eventual lethality. The inclusion of a wash-out recovery step enables animals to survive but could complicate the interpretation that microtubules are necessary for

*wnt1* and *notum* expression behavior, because recovering animals may have unique responses as they eventually re-establish their microtubules during recovery. To examine these possibilities, we tested the importance of the wash-out step in the ability of colchicine to reduce *wnt1* and *notum* expression at 200 µg/ml and elevate *notum* expression at 125 µg/ml. Continuously treating animals with 200 µg/ml colchicine for 24 h pre-amputation and 18 h post-amputation caused a similar reduction to *notum* and *wnt1* expression at anterior-facing wounds as in treatments that included a washout step. Likewise, animals treated with 125 µg/ml colchicine continuously also caused increased *notum* expression at posterior-facing wound sites (Fig. S13). Therefore, microtubule inhibition is the cause of *wnt1* and *notum* expression effects and not as a consequence of recovering from inhibition.

## Microtubules participate with Wnts in head/tail blastema fating

Our model suggested that microtubules may control *wnt1* and *notum* expression phases through parallel pathways. To examine for any participation of microtubules in head/tail decisions that might occur in conjunction with other regulators, we tested whether colchicine could modify the appearance of regeneration phenotypes after Wnt RNAi. RNAi of *wnt1* under these conditions caused a 22% penetrant phenotype of *sfrp-1*+ posterior head formation (5/23 animals), but simultaneous treatment with 125 µg/ml colchicine increased the penetrance to 88% (15/17 animals, *P*<0.0001 by 2-tailed Fisher's exact test) (Fig. 6A). Similarly, *wnt11-2* RNAi animals all failed to regenerate tails without regenerating posterior heads (16/16 animals by live scoring and 28/28 animals lacked posterior *sfrp1* expression), but simultaneous treatment with 125 µg/ml colchicine caused a fraction of these animals to form posterior heads (4/21 animals as measured by posterior *sfrp-1* expression, *P*=0.0432 by two-tailed Fisher's exact test) (Fig. 6A). Therefore, microtubules act with Wnt signaling to regulate the specificity of regeneration. Together, our results indicate that muscle contains a network of microtubules, that microtubules are required for repair of pre-existing muscle fibers across injury sites, that specific microtubule regulatory processes regulate injury-induced expression of *wnt1* and *notum* from muscle, and that microtubules functionally contribute to the *wnt1*-dependent process of head/tail blastema identity determination (Fig. 6B,C).

## DISCUSSION

Our analysis reveals that planarian body-wall muscle harbor a network of microtubules along their actinomyosin-rich contractile fibers and identifies functional roles for microtubules in the process of muscle fiber growth and expression of key patterning genes activated within muscle. By imaging of TUBA-2+ muscle microtubules after incision injuries, we show that body-wall muscle regrows following wounding (Fig. 6B). Because this process occurs in animals irradiated at doses that eliminate stem cells, pre-existing muscle cells are likely the source of repair, rather than differentiation of new muscle cells. Colchicine prevented fiber regrowth, indicating microtubules are crucial for this process. Although microtubules are present broadly in many planarian cell types with a diverse set of functions, we find that microtubule inhibition affects expression strongly for a subset of injury-induced genes. Regeneration polarity factors *wnt1* and *notum* were among the most strongly downregulated genes following microtubule inhibition, indicating that some microtubule-dependent process selectively promotes their expression after injury. At lower colchicine doses, animals underwent a selective upregulation of

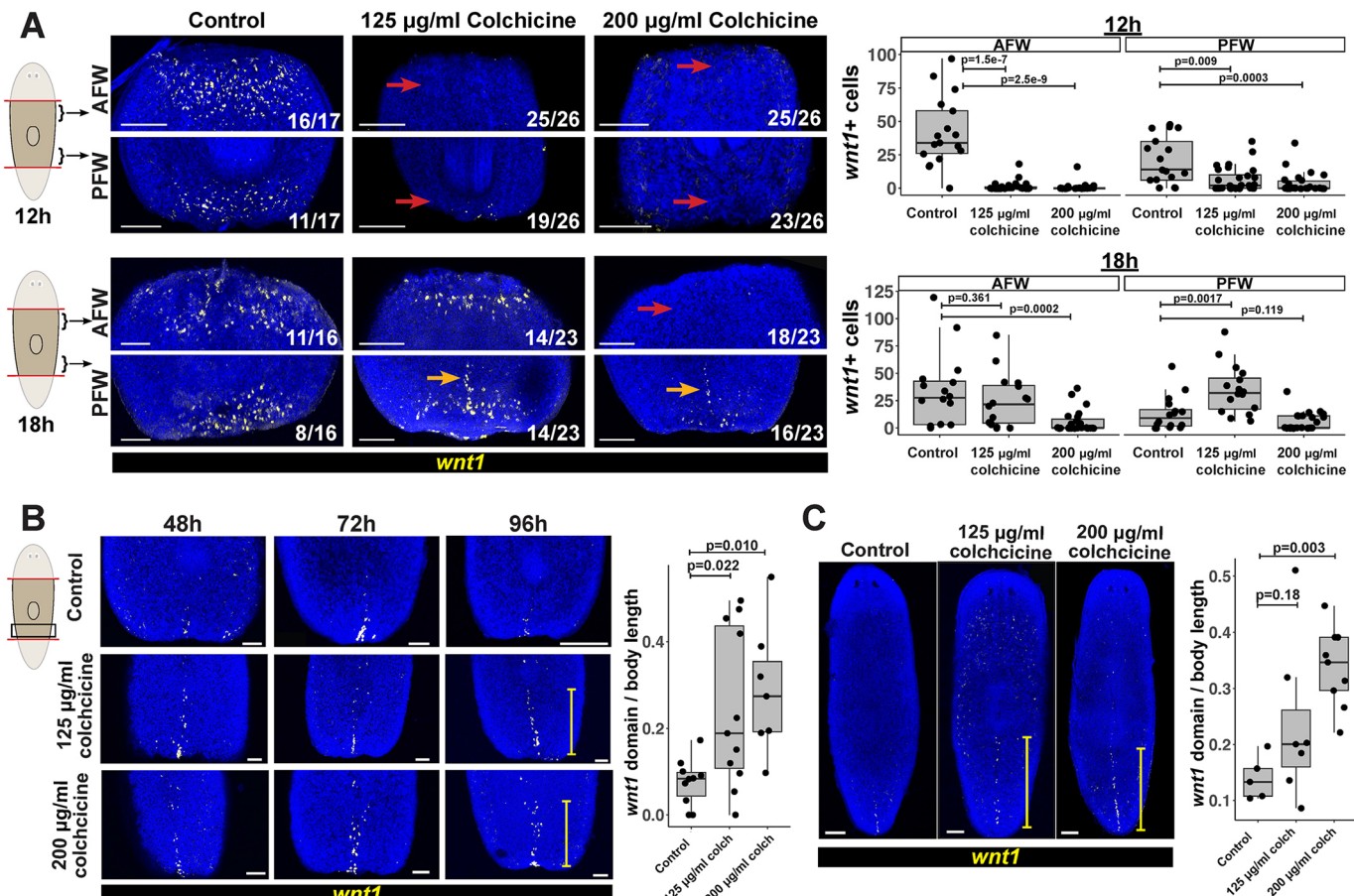

**Fig. 5. Microtubules are required for wound-induced *wnt1* activation and regulate *wnt1* posterior pole expression.** (A-C) FISH detecting *wnt1* following 24 h colchicine pretreatment, amputation, recovery in control media and fixation at indicated times. (A) 200 µg/ml colchicine eliminated *wnt1* expression at anterior- and posterior-facing wounds (red arrows) at 12 and 18 h. 125 µg/ml colchicine delayed the onset of *wnt1* expression to 18 h. Colchicine also caused expression of *wnt1* at the dorsal midline near posterior-facing wounds (orange arrows). Quantifications show *wnt1*+ cells at anterior- and posterior-facing wounds from individual animals (dots), with overlaid boxplots showing median and 25-75% interquartile range. *wnt1*+ scorings include only the non-midline cells observed near the wound site. *P*-values were calculated using Kruskal–Wallis test followed by Dunn's test to compare multiple conditions. (B) Posterior-facing wounds underwent posterior pole *wnt1* expression from 48 to 96 h, even under conditions strongly decreasing the wound-induced phase of *wnt1* at 12-18 h. Right: quantification showing that colchicine increased the length of the *wnt1* expression domain (indicated by the bar on the left) normalized to body length in 96 h regenerating trunk fragments. (C) Uninjured animals treated with 125 or 200 µg/ml colchicine for 3 days underwent anterior expansion of the *wnt1* expression domain (right, quantifications as in B). (C,D) *P*-values were calculated using the Kruskal–Wallis test followed by Dunnett's tests comparing multiple samples against common controls. Scorings show number of animals similar to representative images. Scale bars: 300 µm (A); 50 µm (B,C).

*notum* expression at posterior-facing wound sites in a step requiring *wnt1* and *beta-catenin*, similar to normal expression of *notum* at anterior-facing wounds. Under conditions of strong microtubule inhibition, no *wnt1* expression occurs and therefore *wnt1*-dependent expression of *notum* at anterior-facing wounds does not occur. At lower levels of microtubule inhibition (125 µg/ml), *wnt1* wound-induced expression still activates at posterior-facing wounds, though in a delayed fashion, and the process repressing *notum* at posterior-facing wounds is partially disrupted. This polarity mechanism could be driven by *wnt11-2*, *activin-2* or other as yet unidentified factors. In addition, microtubule inhibition also expanded the *wnt1*+ posterior pole in uninjured animals and also animals regenerating a new tail. In normal animals, the dual sources of *wnt1* from the posterior pole and also early injury activation phases could act in parallel to promote tail versus head blastema fate. This model would explain why dual inhibition of microtubules along with either *wnt1* or *wnt11-2* resulted in increases to the fraction of animals undergoing regeneration with inverted polarity (Fig. 6C).

Our study reveals that microtubules regulate injury-induced genes and control regeneration decision making, and also raises several questions about how microtubules contribute to these processes on a mechanistic level in planarians. Microtubules are known to mutually interact with Wnt signaling through a wide variety of mechanisms across proliferative and post-mitotic cells across many organisms (Kikuchi and Arata, 2024), suggesting myriad opportunities for planarian microtubules to participate in Wnt-mediated control of head versus tail regeneration. In canonical Wnt signaling, Wnt binding to Frizzled receptors activates Disheveled, which inactivates the GSK3 kinase and thereby prevents a destruction complex from constitutively degrading β-catenin protein, which can then undergo nuclear translocation and transcriptional regulation in conjunction with TCF transcription factors. β-Catenin can be transported to the nucleus via microtubules and is dependent on Kinesin-2 and the IFT-A complex (Vuong et al., 2014, 2018; Balmer et al., 2015). APC, a component of the β-catenin destruction complex, is also a microtubule plus-end binding protein and can be transported along

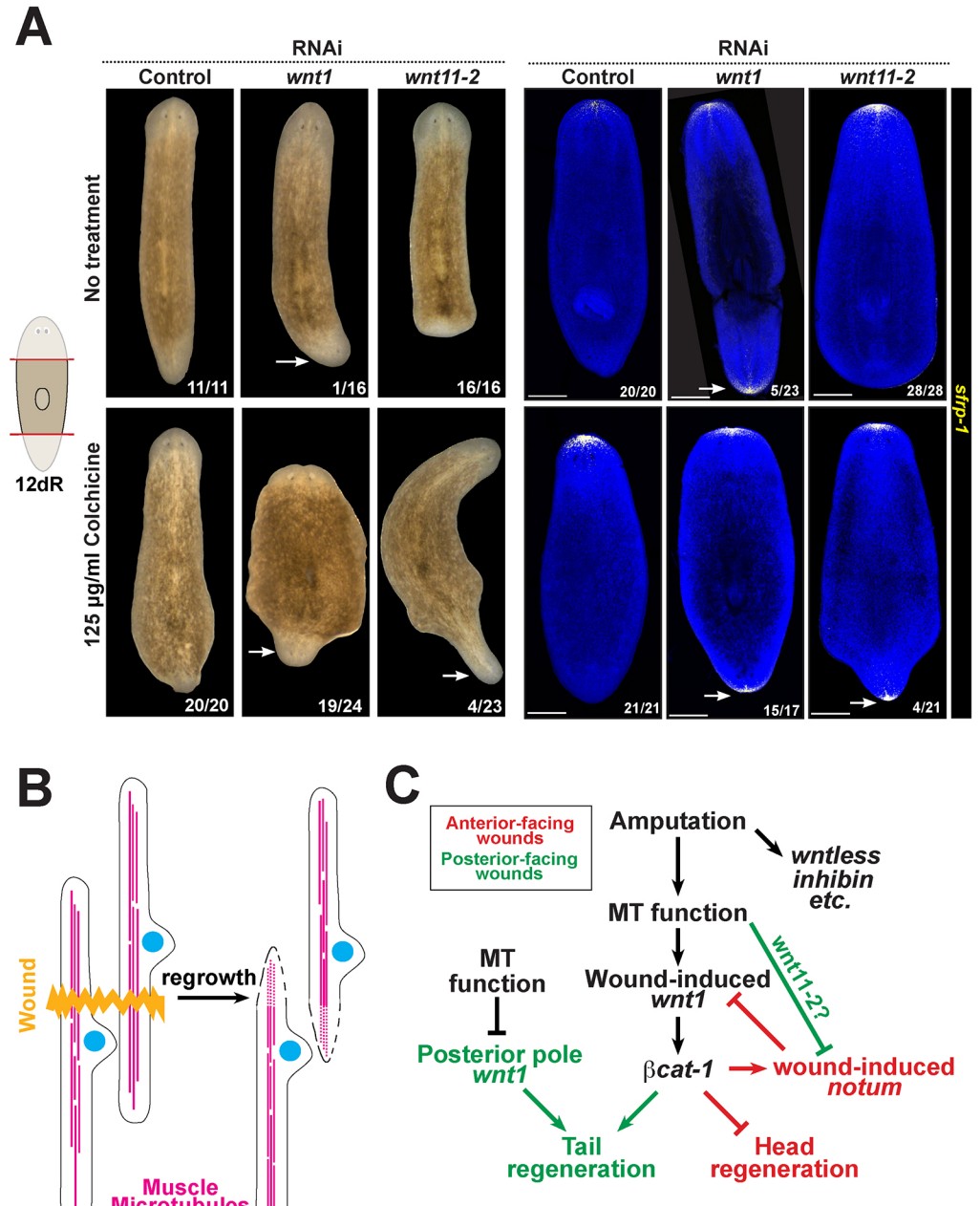

**Fig. 6. Microtubules participate functionally in head-tail blastema determination along with Wnts.** (A) Animals were treated with control, *wnt1* or *wnt11-2* dsRNA for 2 weeks over 6 dsRNA feedings, followed by treatment with or without 125 µg/ml colchicine for 24 h prior to amputation. (Left) Live animals were scored and imaged at day 12, followed by (right) FISH for the anterior marker *sfrp-1*. Scorings represent animals scored as similar representative images. Animals were scored as having inverted polarity if the posterior blastema contained eyes (live images) or contained *sfrp1⁺* cells. Colchicine increased the penetrance of *wnt1(RNAi)* or *wnt11-2(RNAi)* animals regenerating posterior-facing heads. Scale bars: 50 µm. (B) Schematic depicting the organization of microtubules along the muscle fiber with respect to the cell body. Incision injuries result in rapid regrowth of muscle fibers 4-18 h after wounding. (C) Regulatory diagram incorporating the findings from this study. Microtubules promote expression of *wnt1* and *notum*, but not all muscle-expressed injury-induced genes (e.g. *inhibin-1* and *wntless*). *wnt1* and *beta-catenin-1* activate injury-induced *notum*. Strong microtubule inhibition prevents activation of *wnt1* and *notum* at 12-18 h. At posterior-facing wounds and lower doses of colchicine (125 µg/ml), *wnt1* expression is delayed and *notum* expression is overactivated, which may be due to loss of microtubule-dependent tissue polarization involving *wnt11-2* or unknown factors. Microtubule inhibition also leads to expanded *wnt1* expression at the posterior pole, which likely acts in parallel to wound-induced *wnt1* and/or *notum* for regulating tail fates.

microtubules (Jimbo et al., 2002). Wnt signaling components also regulate microtubule dynamics. GSK3, in some cases acting through the Wnt pathway, can regulate microtubule stability and dynamics through phosphorylation and modification of the microtubule plus-ended binding ability of Tau, MAP1B and APC (Lucas et al., 1998; Zumbrunn et al., 2001; Johnson and Stoothoff, 2004; Noble et al., 2005; Caccamo et al., 2007). Axin, a scaffolding protein that functions in the β-catenin destruction complex, binds to γ-tubulin and can be involved in microtubule organizing center formation (Ruan et al., 2012). Dishevelled regulates both canonical and non-canonical Wnt pathways, can participate in the regulation of MAP1B to control microtubule stability (Ciani et al., 2004), and can regulate the orientation of microtubules linked with the cell cortex in migration and cell division (Matsumoto et al., 2010; Yang et al., 2014; Kikuchi et al., 2018). Furthermore, planar cell polarity pathways that involve regulation through Dishevelled and the asymmetric localization

of Frizzled and Vangl cell-surface proteins can rely on microtubules for polarized transport of these proteins within the cell (Shimada et al., 2006; Matis et al., 2014), and downstream outcomes of planar cell polarity can also be regulated through control of microtubule dynamics (Vladar et al., 2012). Therefore, the function of microtubules to support wound-induced expression of *wnt1*, the regulation of *notum* polarity and the control of head-versus-tail blastema specification could, in principle, arise from these or other potential relationships. The analysis of microtubule regulatory factors and also the generation of new tools to allow assessment of Wnt pathway component localization in planarians will be an important future step to further understand these mechanistic links.

Based on our phenotypic analysis, our work rules out several different candidate models for how microtubules promote injury-induced expression of *wnt1* and *notum*. Muscle from injured animals treated with 200 µg/ml colchicine could still activate expression of

several wound-induced genes, such as *nlg1*, *wntless* and *inhibin*, and these animals still had abundant *collagen*[+] and *myoD*[+] muscles, so the lack of *wnt1* activation in these animals is unlikely to be due to a nonspecific requirement for microtubules to maintain muscle itself or its general wound responsiveness. Our RNAseq analysis to identify genes differentially expressed following colchicine treatment in planarians also did not detect modifications to expression of the *wnt1*-inducing transcription factors *foxG* or *arx-3* (Akheralie et al., 2023; Pascual-Carreras et al., 2023). Additionally, *wnt1* expression does not require *beta-catenin-1* (Petersen and Reddien, 2009), so microtubule-mediated regulation of *wnt1* occurs likely through another pathway. Planarian *hedgehog* is necessary for *wnt1* expression (Rink et al., 2009) and does not control *notum* expression polarization (Petersen and Reddien, 2011). Hedgehog signaling involves Kif7-mediated microtubule transport within the primary cilium in vertebrate cells (He et al., 2014), although earlier analyses have not found support for Cos2/Kif27/Kif7 kinesins being involved in the transduction of Hedgehog signaling effects relating to head/tail determination in planarians, but rather in Hedgehog-independent ciliogenesis (Rink et al., 2009). However, our experiments do not rule out the possibility that a variant Hedgehog pathway could be relevant for microtubule activity and *wnt1* expression in muscle. In addition, microtubules could regulate *wnt1* through *follistatin*, which is induced from muscle cells at injury sites and negatively regulates injury-induced *wnt1* expression independent of its role to enable the activation of proliferation and cell death in an animal following large injuries (Tewari et al., 2018). However, wound-induced *follistatin* expression still occurred but at reduced levels following colchicine treatments that, nonetheless, eliminated *wnt1* and *notum* expression. These observations are inconsistent with a hypothetical model in which microtubules suppress *follistatin* activation in order to limit *wnt1* activation. Microtubule action in other types of muscle could also be important for *wnt1* expression. The expression of *wnt1* on the posterior dorsal midline is negatively regulated by STRIPAK components *mob4* and *striatin* (Schad and Petersen, 2020), *bmp* (Clark and Petersen, 2023), and the kinase *pak1* acting upstream of *warts* and *yorkie* in the Hippo pathway (Lin and Pearson, 2014, 2017; Doddihal et al., 2024). Microtubules similarly restrict this *wnt1* domain posteriorly within the tail, suggesting possible uses for microtubules upstream or downstream of these factors. Future work will be necessary to resolve how microtubules regulate the *wnt1*-related pathways that together impinge on head/tail determination.

Likewise, several candidate models could potentially explain how microtubules are involved in the polarization of *notum* expression. One possibility is that microtubules are responsible for asymmetric transport or sequestration of mRNA/protein polarity factors along muscle fibers, such that these cells are primed for either activation or suppression of *notum*, depending on their orientation to a wound site. Because *notum* is a feedback inhibitor of β-catenin signaling, it is also possible that known mechanisms relating microtubule regulation to canonical or noncanonical Wnt signaling could be responsible for the role of microtubules in suppressing *notum* expression specifically at posterior-facing wounds (described above and reviewed by Kikuchi and Arata, 2024). Alternatively, colchicine could disrupt planar cell polarity signals that emerge from polarized tissues, such as the planarian epidermis (Almuedo-Castillo et al., 2011; Vu et al., 2019), that generate polarized *notum* expression outcomes within nearby muscle at injury sites; however it unknown whether epidermis is necessary for *notum* polarization. Microtubules are intrinsically polarized polymers whose organization within muscle could itself imbues these cells with polarity. Microtubules could either be oriented in a consistent direction within each type of muscle aligned

with the body axis (e.g. plus-ends anterior in all longitudinal muscle) or such alignments could occur in each cell but not uniformly for all muscle cells (e.g. all longitudinal muscle has polarized microtubules but they adopt random orientations). Alternatively, they could adopt a distribution of orientations within each fiber (i.e. plus ends of individual tubules aligned in either direction along the fiber) or have a common orientation with respect to the cell body (e.g. minus ends all located at the cell body and plus ends distally in the fibers). Although no A/P differences in individual muscle fibers are known, light-sheet imaging of 6G10-labeled muscle found a distinct geometry of fiber alignment at the anterior versus posterior termini, and the fiber network morphology was overall modified by *beta-catenin-1* RNAi, suggesting that the muscle network structure is responsive to head-tail patterning (Lu et al., 2025). The establishment of additional reagents for detection of microtubule end proteins and microtubule-associated factors will be important for resolving how microtubules are oriented in planarian muscle to further understand how they may contribute to polarized blastema choices.

Our study demonstrates that microtubules functionally participate in conjunction with Wnt pathway components to control regeneration decisions in planarians. This participation could include both promoting injury-induced *wnt1* expression and suppressing *notum* at posterior-facing wounds, as both would bias blastema specification toward tail fates. In addition, we find microtubule inhibition does not eliminate, but actually expands, *wnt1* expression at the posterior pole. Our findings that colchicine treatment dramatically enhanced the *wnt1(RNAi)* and *wnt11-2(RNAi)* phenotypes could be consistent with these three pathway branches acting in series, given likely incomplete inhibition from RNAi and drug treatments, or acting in parallel. Consistent with this idea, while *notum* expression asymmetry is prominent in some planarians, i.e. *Schmidtea mediterranea* (Petersen and Reddien, 2011) and *Dugesia japonica* (Durant et al., 2019), its expression is symmetric in *Girardia sinesis*, which undergoes whole-body regeneration but with naturally frequent head/tail polarity reversals (Cleland et al., 2025 preprint). Therefore, while *notum* asymmetry is not a universal feature of planarian regeneration, it may facilitate robust blastema decision making. Our analysis suggests the head/tail decision process involves multiple redundant events that could vary across species and help ensure regeneration robustness in *S. mediterranea.*

How damage signals are integrated to produce specific regeneration outcomes is still not well understood at the cellular level in any organism. Our study raises the possibility that damage to muscle fibers provides a signal that activates expression of a set of genes that subsequently regulate regeneration decisions. Future work will be crucial to uncover the triggers and relay mechanisms central to this process. The use of microtubules in actinomycin-rich cell projections could have a more general purpose in mediating injury responses. Our analysis identifies a network of microtubules in planarian muscle and defines microtubules as crucial for regulating specific injury-induced programs involved in the regulation of blastema identity.

## METHODS AND MATERIALS
### Experimental model
Asexual *Schmidtea mediterranea* (CIW4 strain) were cultured in 1× Montjuic salts at 18-20°C. Animals were fed pureed calf liver once a week and cleaned at least once every week in static cultures for maintenance. Animals were starved for at least 7 days before experiments.

### RNAi
RNAi treatments were performed by feeding with dsRNA (16% v/v) at a final concentration of ~300 ng/μl, green food dye (4% v/v)

and pureed liver (80% v/v). Animal of each condition were cultured in separate Petri dishes in 1× Montjuic salts. dsRNA was synthesized as previously described (Bonar et al., 2022; Clark and Petersen, 2023). RNAi negative controls used dsRNA targeting *C. elegans unc-22*, a gene not present in the *Schmidtea mediterranea* genome. Animals were fed 10 µl of RNAi food mixture per worm every 2-3 days for the indicated length of the experiment. For RNAi treatment without injury, animals were fixed 5 days after the final feeding. For a regeneration time courses following injury, surgery was performed 2 days after the final feeding, prior to fixations at the indicated times.

### Microtubule inhibition
For preparation of colchicine-containing media, 500 µg/ml colchicine (Sigma C9754-500MG) dissolved in 1× Montjuic salts was prepared freshly and then diluted to desired concentrations in 1× Montjuic salts. Animal media were replaced with colchicine-containing media for 24 h and, if surgeries were performed, animals were allowed to recover in colchicine-free media while they regenerated until further processing. For experiments involving colchicine, planarian water (1× Montjuic salts) was used as a negative control. Nocodazole (Sigma 487929-10MG-M) was dissolved in 1% DMSO in 1× Montjuic salts at the indicated concentrations and the solution made freshly before use. Animal media were replaced with nocodazole-containing media for 24 h at the indicated concentrations, then if surgeries were performed, animals were allowed to recover in 1× Montjuic salts without nocodazole for the indicated times prior to fixation and analysis. For experiments involving nocodazole, mock treatment of animals with 1% DMSO in planarian water were used as negative controls.

### Fluorescence *in situ* hybridization
Fluorescence *in situ* hybridization (FISH) was performed using established protocols (King and Newmark, 2013). Briefly, animals were treated with 7.5% (w/v) N-acetyl cysteine (NAC, Sigma A7250-100G) in 1×PBS, fixed in 4% formaldehyde (w/v) (Sigma F8775-500ML) and stored in methanol (ThermoFisher A4544). They were rehydrated with methanol: 1× PBSTx (PBS with 0.3% Triton X-100 Sigma, T8787-250ML), bleached in 1.2% $H_2O_2$ (v/v) (H1009-500ML), 5% formamide and 0.5xSSC in 1× PBS, permeabilized with proteinase K (10 µg/ml; Invitrogen, 25530049), then prehybridized at 56°C in pre-hybridization hybridization buffer (50% formamide, 5×SSC, 1 µg/ml yeast RNA, 0.5% Tween). Animals were incubated for 16 h in hybridization buffer (50% formamide, 5×SSC, 1 µg/ml yeast RNA, 0.5% Tween and 5% dextran sulfate) containing 1:1000 riboprobes, followed by two washes each of prehybridization buffer, then in 2×SSC/0.1% Triton-X, and 0.2×SSC/0.1% Triton-X. Hybridization occurred with digoxigenin- or fluorescein-labeled riboprobes at a 1:1000 concentration (v/v), which were synthesized using T7 RNA-binding sites for antisense transcription. Anti-digoxigenin-POD (Roche/Sigma 11207733910) or anti-fluorescein-POD antibodies (Roche/Sigma 11426346910) were in a solution of 1× TNTx/10% (v/v) horse serum (Sigma, H1138-500ML)/10% (v/v) Western Blocking Reagent (Roche 11921673001) at a concentration of 1:2000 (v/v). Homemade fluorescein tyramide or rhodamine tyramide prepared as described previously (King and Newmark, 2013) was used in TSA buffer (2 M NaCl and 0.1 M Boric acid and pH 8.5) for 10 min to label the specimens, followed by seven washes in TNTx. For double FISH, the enzymatic activity of tyramide reactions was inhibited by sodium azide (100 mM). Nuclei were stained using 1:1000 Hoechst (v/v, ThermoFisher H3570) in 1×TNTx.

### Immunostaining
Anti-TUBA-2 antibody was generated by Genscript using a C-terminal peptide derived from the residues 440-454 of the TUBA-2 (dd508, SMEST054547001.1) protein sequence with an N-terminal cysteine appended for KLH conjugation (N-CVGYDSADIGNADQD-C). Blast searching confirmed no other exact matches to this peptide exist in the planarian proteome. For immunostaining, animals were fixed in 10 ml glass scintillation vials by replacing planaria water with 2% HCl in water for 30 s with gentle swirling, then media were gently replaced with Carnoy's solution (containing 60% ethanol, 30% chloroform and 10% glacial acetic acid) without disturbing specimens, incubated without mixing for 2 h on ice (4°C), followed by incubation in methanol at −20° for at least 1 h. Animals were bleached in 6% hydrogen peroxide (H1009-500ML) in methanol (v/v) (ThermoFisher A4544) overnight on a lightbox prior to immunostaining. For immunostaining with anti-TUBA-2 (this study) and 6G10 (Developmental Studies Hybridoma Bank 6G10-2C7) antibodies, animals were blocked with 10% horse serum (Sigma H1138-500ML) and 10% Roche Western Blocking Reagent (Roche 11921673001) in PBSTx then incubated overnight at room temperature with primary antibody (anti-TUBA-2 polyclonal rabbit antibody at 1:1000, 6G10 mouse monoclonal antibody at 1:1000) in blocking solution. For H3P staining, animals were incubated in 5% horse serum in PBSTx then incubated overnight at room temperature with rabbit anti-ser10-H3P antibody (Cell Signaling Technology, 3377S) at 1:300 diluted in blocking solution. Animals were washed with PBSTx six times over 6 h prior to incubation with secondary antibodies. Secondary antibodies and concentrations were: goat-anti-rabbit-alexa568 (ThermoFisher A11036) at 1:1000 and goat-anti-mouse-Alexa488 (ThermoFisher A32723) at 1:1000 in Fig. 1A-D,G; goat anti-rabbit-HRP (ThermoFisher G21234) at 1:1000 and goat anti-mouse-alexa568 (ThermoFisher A11031) at 1:1000 in Figs 1F, 2, Figs S1C, S3; or goat anti-rabbit-HRP (ThermoFisher G21234) at 1:1000 in Fig. S2. Tyramide development in 1× PBSTi (PBSTx with 10 mM Imidazole) was performed as described previously (Pearson et al., 2009) to label samples stained with HRP-conjugated secondary antibodies. For nuclear counterstain labeling, samples were incubated in Hoechst dye at 1:1000 and washed at least four times prior to mounting in Vectashield (Vector Laboratories, H-1000) and imaging. Experiments to perform FISH and immunostaining followed a NAFA (nitric acid and formic acid) fixation and staining protocol described previously (Guerrero-Hernandez et al., 2024). Briefly, animals were fixed in NA (nitric acid) solution containing 4% paraformaldehyde (Electron Microscopy Services 15710) in 100 mM HEPES (pH 7.5), 25 mM EGTA, 50 mM MgSO4 and 0.53% nitric acid) for 1-2 min, followed by FA (formic acid) solution [4% paraformaldehyde, 100 mM HEPES (pH 7.5), 25 mM EGTA and 4.80% formic acid (Sigma F0507-500ML)] for 45 min, washed twice in 1× PBS, once in 50% methanol/1×PBS and then transferred into 100% methanol, incubated in 100% methanol for at least 1 h at −20°C, followed by transfer in to 1×PBS and bleaching in formamide bleach solution for 2 h under a light source. Tyramide-FISH detection proceeded as above described in the FISH procedures but omitted the proteinase K step as in the published NAFA-FISH protocol (Guerrero-Hernandez et al., 2024). Samples were then detected by immunostaining, using 1:1000 anti-TUBA-2 primary antibody, followed by 1:1000 anti-rabbit-488 as described above. Samples were labeled with 1:1000 Hoechst dye and washed six times prior to mounting and imaging.

## Cloning

Genes were cloned by RT-PCR after cDNA synthesis (Superscript III, oligo-dT priming) from mixed stage planarian total RNA. Primers used for cloning *wnt1*, *wntP-2*, *wnt11-2*, *runt-1*, *beta-catenin*, *bmp* and *slit* have been described previously (Petersen and Reddien, 2008, 2009; Wenemoser et al., 2012; Lander and Petersen, 2016; Cloutier et al., 2021) or were as follows: *nlg-1* (CGAGAACCGTTGATAG-TTAATGC, CAGCTACATGTGCAAGATTCAT), *wntless* (TCG-ATTGGATGGAGATGAGGT, AACTCCTTCGATGATGCCGT), *inhibin-1* (TGTTACAATGTAGCAGTTGCCA, TCGTCTTTGC-ACTTCAAGAGGA) and *tuba-2* (TCCGCATGTGTCTTTTGGAA, GCCAAATCTTCACGAGCCTC). PCR-mediated addition of T7 sequences was used to generate templates for riboprobe and dsRNA synthesis.

## Irradiation

Animals were irradiated using a Radsource RS-2000 X-ray to deliver 6000 rads to animals in 1× Montjuic salts (Fig. 2B), a treatment that is known to eliminate stem cells (Vasquez-Doorman and Petersen, 2014), and flank incisions were performed 3 days later.

## Cell quantification

For experiments involving quantification of *notum*, *wnt1* or *follistatin*[+] cells, maximum projection images from ~200 µm regions near the wound sites were manually scored by tabulation using Fiji's cell counting plugin. For measurements of *wnt1*, *wnt11-2* and *wntP-2* expression domains, relative lengths of gene expression domains from the posterior tip were normalized to body length in Fiji. H3P[+] cells and animal areas were counted using an automated implementation of Analyze Particles function. Boxplots with overlayed jittered dotplots were generated using shiny.chemgrid.org/boxplotr or ggplot in R. Statistical analyses are described in each figure legends and calculated in R. For comparing multiple samples across a single variable if data were normally distributed (Shapiro's test) and of equal variance (Levene's test), one-way ANOVA followed by Tukey's post-hoc test was used. For data not normally distributed and/or having unequal variance, and in which multiple samples were compared to a common control condition, Kruskal–Wallis non-parametric tests followed by Dunnett's post-hoc tests were applied. For data not normally distributed and/or having unequal variance, and in which multiple samples were compared to each other, Kruskal–Wallis non-parametric tests followed by Dunn's post-hoc tests were applied. Comparisons with $P < 0.05$ were considered significant.

## Gene expression profiling

Animals were treated with control media or 200 µg/ml colchicine for 24 h, followed by head amputation and recovery in colchicine-free media for 0, 4 and 18 h prior to isolation of total RNA from wound-proximal tissue. Samples were derived from 10 animals per biological replicate over 4 biological replicates per condition. Animal fragments were placed in Trizol then homogenized using a Turrex tissue homogenizer, followed by extraction of total RNA. cDNA libraries were prepared and sequenced by Novogene using a directional eukaryotic mRNA expression profiling protocol, which involved capture of mRNA on oligo-dT magnetic beads, reverse transcription using random hexamers, followed by second-strand synthesis, adaptor ligation and paired-end Illumina sequencing to a depth of 30 M reads per sample. Reads were mapped to the Dresden ddv6 transcriptome (https://planmine.mpinat.mpg.de/planmine/begin.do) using HISAT2, followed by differential expression analysis with DESeq2 and using the Benjamini-Hochberg method to correct *P*-values for false-discovery. Heatmaps were created through Rstudio to determine z-scores of log2 (FPKM+1) values of each gene across the treatments. Venn diagrams were constructed in R studio the gplots and ggplot2 packages.

## Acknowledgements
We thank members of the Petersen lab, Dr S. Wignall and Dr V. Gelfand for helpful advice and discussions, and B. Stevens for help with automated H3P cell counting.

## Competing interests
The authors declare no competing or financial interests.

## Author contributions
Conceptualization: C.P.P.; Funding acquisition: C.P.P.; Investigation: X.N.A.; Project administration: C.P.P.; Supervision: C.P.P.; Writing – original draft: X.N.A., C.P.P.; Writing – review & editing: X.N.A., C.P.P.

## Funding
This work was supported by the National Institutes of Health (NIGMS R35GM149280 and 5F31GM149178). The funders had no role in study design, data collection and analysis, decision to publish, or preparation of the manuscript. Open Access funding provided by Northwestern University. Deposited in PMC for immediate release.

## Data and resource availability
RNAseq datasets have been deposited in NCBI under accession number PRJNA1212034. All other relevant data and details of resources can be found within the article and its supplementary information.

## Special Issue
This article is part of the Special Issue 'Lifelong Development: the Maintenance, Regeneration and Plasticity of Tissues', edited by Meritxell Huch and Mansi Srivastava. See related articles at https://journals.biologists.com/dev/issue/152/20.

## Peer review history
The peer review history is available online at https://journals.biologists.com/dev/lookup/doi/10.1242/dev.204669.reviewer-comments.pdf

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
