## [Peer Review File · Development (Cambridge, England)]

Planarian microtubules form a network within muscle and regulate injury-induced genes essential for regeneration patterning

Xavier N. Anderson and Christian P. Petersen

DOI: 10.1242/dev.204669

Editor: Kenneth Poss

Review timeline

Original submission:	21 January 2025
Editorial decision:	28 February 2025
First revision received:	20 August 2025
Accepted:	17 September 2025

Original submission

First decision letter

MS ID#: dev.204669

MS TITLE: Planarian microtubules form a network within muscle and regulate injury-induced genes essential for regeneration patterning

AUTHORS: Xavier Anderson; Christian P. Petersen

Dear Chris,

I have now received all the referees' reports on the above manuscript, and have reached a decision. The referees' comments are appended below, or you can access them online: please go to:

As you will see, the referees express considerable interest in your work, but have some significant criticisms and recommend a substantial revision of your manuscript before we can consider publication. For example there are concerns about the specificity of the effects of pharmacological microtubule disruption, and on the clarity of mechanism and conceptual advancements. If you are able to revise the manuscript along the lines suggested, which may involve further experiments, I will be happy receive a revised version of the manuscript. From reading the comments, I feel that pulldown of Tub3a and proteomics as suggested by Reviewer #3 is a good experiment but not necessary for a revision you might prepare. Your revised paper will be re-reviewed by one or more of the original referees, and acceptance of your manuscript will depend on your addressing satisfactorily the reviewers' major concerns. Please also note that Development will normally permit only one round of major revision. If it would be helpful, you are welcome to contact us to discuss your revision in greater detail. Please send us a point-by-point response indicating your plans for addressing the referees' comments, and we will look over this and provide further guidance.

Please attend to all of the reviewers' comments and ensure that you clearly highlight all changes made in the revised manuscript. Please avoid using 'Tracked changes' in Word files as these are lost in PDF conversion. I should be grateful if you would also provide a point-by-point response detailing how you have dealt with the points raised by the reviewers in the 'Response to Reviewers' box. If you do not agree with any of their criticisms or suggestions please explain clearly why this is so.

Reviewer 1

SUMMARY OF THE ADVANCE MADE IN THIS PAPER AND ITS POTENTIAL SIGNIFICANCE TO THE FIELD

Anderson et al. studied the formation of a microtubule network within planarian muscle and its role in regulating injury-induced genes crucial for regeneration patterning. The specific staining of microtubules in body-wall muscles, which can be modulated by drugs targeting microtubules, is particularly noteworthy. This study used this valuable tool to elucidate how the asymmetry of body polarity is achieved through the longitudinal muscles. The findings also provide insights into the asymmetric expression of notum after injury in planarians. I believe it would be significant to show how the longitudinal muscles align across the entire animal to understand the regulation of notum expression across the scale of the entire animal, given the relatively short length of the longitudinal muscle fibers. A more in-depth focus on this aspect in the discussion would facilitate understanding. My major and minor comments are as follows.

SUGGESTIONS TO AUTHORS

1. The most significant conclusion I seek is an understanding of how microtubules within muscle fibers regulate cell polarity and the asymmetric distribution of wnt components. The authors highlight that longitudinal muscle fibers are ~ 200 μm in length, presenting a challenge in understanding the regulation of wnt components throughout the entire planarian body. Alternatively, there must be cell-cell connections to facilitate alignment through the body. I would appreciate visual evidence of the longitudinal muscle fibers to determine whether a single muscle fiber extends from the head to tail, or from the anterior wound to the posterior wound.
2. I believe that Figure 1 and 2 could be merged into a single figure. It would allow the text to be more effective in the functional discovery, as both figures are intended to demonstrate that the antibody targets the microtubules within the muscle cells. Similarly, Figure 4 and 5 would also be better to be combined into one figure.
3. In Figure 7, the results indicate that wnt1 RNAi combined with Colchicine treatment increased the penetrance of posterior head regeneration. However, Figure 6, demonstrates that wnt1 RNAi diminishes the posterior expression of notum when treated with 125 $\mu\text{g}/\text{mL}$ colchicine. How can these results be interpreted and integrated into the pathway model? At this concentration of colchicine, reduced activin-2 and sustained wnt1 may lead to the expression of notum posteriorly. However, wnt1 RNAi leads to a reduction in posteriorly expressed notum, complicating the understanding of how these results align with the proposed pathway.
4. It is intriguing to consider how the morphology at the flank incision wound site was preserved during the fixation and staining procedures. Technically, this understanding is critical for accurately interpreting the results.
5. The transition from the suppression of notum expression to the ectopic expression at the posterior wound is "surprising". It would be beneficial to provide additional data to support this observation, as all the RNAseq data analysis are from the 200 $\mu\text{g}/\text{mL}$ Colchicine treatment. If the recovery of microtubules during regeneration explains the normal regeneration outcome, it is essential to specify the days post injury when the microtubule recovered to normal levels. For instance, at 5 dpa the authors showed a delay in regeneration after treatment of colchicine for 24 hours and subsequent regenerate in control media. Have the microtubules recovered at this time point, and what is the expression of notum at the posterior pole?
6. In lines 359 and 362, the referenced figures should be Figure 6A and 6B.
7. Figure 1D shows the muscle cell body surrounded by TUB3A proteins. Conversely, I did not observe similar patterns in Figure 1E. Is this difference due to the two staining methods or is it merely a result of not displaying the images in a similar manner.?
8. The image resolution in Figure 2A is inferior to that in Figure 1. Please address this issue. How many animals were used as controls in Figure 2B? Additionally, interpreting the black upper right

corner in the right panel of Figure 2B for 6G10 and TUB3A co-staining is challenging. Were there anything issues following the laser bleach?

9. What is the change in expression of activin-2 after colchicine treatment at 125µg/mL? The data presented in figureS10 are for 200µg/mL, not 125µg/mL.

10. Determining whether notum is the only gene increased in the PFW may be beyond the scope of this study. However, it is quite interesting to conduct RNAseq on the PFW tissues to confirm the decrease in wnt1 and the increase in notum.

11. In the Methods Section, additional details are required. please specify whether the worms are maintained in a circulation system. The dsRNA concentration needs to be specified. The catalog number of each reagent should be provided. Additionally, clarify what is PBSTi represents.

12. Due to the unavailability of the Tables for verification, I have not confirmed this part of data.

Minor points:

1. There are a couple of corrections in the References:

Please ensure the correct use of the Latin alphabet in authors' name. For instance, "Sanchez" should be corrected to "Sánchez", and "SALO" should be corrected to "SALÓ". Please make these adjustments throughout the references.

Line 1125 and 1128, the authors cited the same paper as two references. Please ensure the authors' names are formatted consistently. For example, ALVARADO, A.S., or SANCHEZ ALVARADO, A.

2. Understanding whether the anti-TUB3A staining exhibits distinct staining signals under the two different fixation protocols is beneficial.

3. The scale bars of Figure 1E are missing.

4. Is Nocodazole media also prepared freshly, similar to the colchicine media?

5. When authors claim the "muscle cell body could be a site of tubulin monomer accumulation", please clarify that colchicine is to inhibit the tubulin heterodimer from polymerization. It should be noted that this is a site of tubulin heterodimer accumulation.

6. The thickness of the scale bars should be consistently maintained in Figures.

7. Please adjust the font of µg in figures.

8. Muscle fiber regrowth across wound sites has recently been reported in Lu et al., eLife, 2024, which can be considered to be included in the citation.

9. TUB3A or Tub3A should be consistent in Figure 3 and throughout the manuscript.

10. H3P cells should be H3P+ cells. mm² should be mm².

11. A 2-tailed unpaired t-test should be referred to as a two-tailed unpaired t test. Is this Student's t test? If so, please specify accordingly.

12. In the legend of Figure S5, there is a redundant description of runt-1 expression. Please remove one instance.

13. On line 614, a citation is labeled as [29], which is not consistent with other citations.

Reviewer 2

Review Manuscript: dev.204669, Anderson & Petersen

During regeneration, cells must undergo dramatic and dynamic changes in cell shape and identity. Though the cytoskeleton has long been known to regulate diverse cellular processes, including cell migration and cell shape, it has not been well characterized in whole-animal regeneration processes. In this paper, Anderson and Petersen have investigated specific properties of the microtubule cytoskeleton in planarian flatworms, exploring the role of microtubules in muscle cells in regeneration. Importantly, muscle cells in planarians have typical contractile roles in animal movement and wound healing, but they also serve as critical signaling sources of cues necessary to repattern animal tissue after amputation. This paper fills a much needed gap in our understanding of the cellular biology of muscle cells in planarians and in the role of the cytoskeleton in driving specific aspects of regenerative biology. Strengths of the manuscript include: 1) its innovative focus on cell type-specific cytoskeletal proteins to shed light on microtubule biology in planarian muscles; 2) a careful return to and replication of prior experiments perturbing microtubules using chemical inhibitors; and 3) rigorously quantified data (including a fully validated new antibody). One weakness is the lack of clear mechanism connecting microtubules to polarity signaling. This weakness is mitigated to a large extent for overall enthusiasm about this exciting project. This paper is a strong fit for Development, though a number of concerns could be addressed prior to publication.

Major:

How colchicine affects muscle cell biology and animal-level regeneration was a bit underdeveloped. More information on the overall animal response to colchicine and how cells are impacted could help shed light on the results with Notum and polarity.

1. Are muscles able to contract sufficiently to allow timely wound closure after lateral cut or amputation injuries? Some figures like S3 show features that might be interpreted as slow wound closure.
2. Is wound response (in terms of gene expression) changed temporally after colchicine treatment. Some of the results in Fig. 4 might be consistent with a delayed or prolonged wound response, which would likely change the reader's interpretation of these results.
3. Are small blastemas seen after colchicine treatment and regeneration? For pieces with a short A/P length, do any blastema defects (e.g. split blastema) or reversed polarity phenotypes occur as was seen in McWhinnie, 1955? Do these require continuous colchicine administration or is the 24 hour period enough?
4. Is posterior signaling (e.g. wnts) normal at 12dR after colchicine treatment? The tail looks a bit underdeveloped in the live animal image.
5. What is happening to muscle cell shape and connectivity after colchicine treatment in the presence of injury? The authors note decreased 6G10 staining. Are other antibodies available to stain MHC (e.g. TMUS-13 used by F. Cebrià, 2000) to take a look at actomyosin? Can cells be dissociated to see if muscle cells maintain their shape or length (as per Witchley, 2014)? These experiments would help to understand exactly the impact of microtubules on muscle cells.

Minor:

1. (Optional) I found myself wondering more about the genes differentially expressed at the 0h time point. This could be addressed in a supplemental figure for others in the field who are interested. In which cells were these genes expressed (predominantly muscle or a mix? What categories (GO terms, for example) are overrepresented? I think this is worth including so that others can build on this interesting work!
2. There are some statistical errors:
 - a. t-tests are not appropriate for experiments with 3 samples (e.g. Fig. 5 and similar). Instead, one-way ANOVA with a post hoc pairwise test would be appropriate for this type of experiment.
 - b. t-tests are not appropriate for experiments with 2 variables (e.g. Fig. 6 and similar). Instead, a two-way ANOVA would be more appropriate.
3. Occasionally, the text indicates that cytoskeletal filaments are being observed rather than cells.
 - a. Line 74, "Planarian muscle cells possess actinomyosin contractile fibers that project...." It is the muscle fibers consisting of end-to-end cells rather than the contractile fibers that project across the body.
 - b. Line 236, "TUB3A+ microtubules could be observed..." It is probably the muscle cells being observed at this magnification. (See also lines 238 and 240).

4. Using magenta and green rather than red and green can help readers with color vision to see the beautiful staining data better.
5. There is a typo in regeneration in Fig. 8.

Reviewer 3

The manuscript by Anderson et al., titled "Planarian Microtubules Form a Network Within the Muscle and Regulate Injury-Induced Genes Essential for Regeneration Patterning," explores the role of microtubules in regulating the expression of wound-induced genes, specifically *wnt-1* and *notum*, which are critical for establishing anterior-posterior (A/P) polarity during regeneration. The authors identify a muscle-expressed tubulin isoform, Tub3A, but its knockdown does not result in a discernible phenotype. Nevertheless, the authors attribute the microtubule function to colchicine treatment, which depolymerizes microtubules, leading to the downregulation of muscle-expressed, wound-induced genes and resulting in defective A/P polarity. Although the authors present several important observations, many of their conclusions are based on correlational evidence and remain highly speculative. The manuscript lacks a clear mechanistic explanation for how microtubules regulate wound-induced gene expression.

Major Comments:

1. The authors use different concentrations of colchicine to perturb microtubule polymerization. Since microtubules are essential for muscle function and integrity, and are also involved in the transport of proteins and RNA, it is not surprising that perturbing such a crucial molecule could have long-lasting effects on muscle integrity. The phenotypes observed could, therefore, be due to disrupted muscle integrity rather than a direct role of microtubules in regulating *Wnt-1* and *Notum* expression.
2. The authors show that 200 $\mu\text{g}/\text{ml}$ colchicine treatment causes more global defects such as cell cycle defects etc, whereas the 125 $\mu\text{g}/\text{ml}$ colchicine concentration seems to affect muscle more specifically. However, it can be argued that 125 $\mu\text{g}/\text{ml}$ colchicine might perturb specific muscle types more than others. To address this, the authors should examine the expression of other muscle-specific markers, such as *myoD* and *nkx1-1*, to assess overall muscle integrity. Furthermore, the authors should also test the expression of additional polarity genes, such as *BMP* and *Slit*, which mark dorsal/ventral (D/V) and midline regions. Perturbation of microtubules might also affect these genes, in addition to A/P markers like *Wnt-1* and *Notum*.
3. I am surprised that transcriptome sequencing was not conducted following treatment with 125 $\mu\text{g}/\text{ml}$ colchicine. I recommend performing transcriptome sequencing for both 125 $\mu\text{g}/\text{ml}$ and 200 $\mu\text{g}/\text{ml}$ colchicine treatments and comparing the results. This would allow identification of specific polarity markers and muscle types that are more dramatically affected by each treatment.
4. Since the authors have raised an antibody against the muscle-specific tubulin, Tub3A, it would be valuable to perform pull-down assays with this antibody followed by proteomic analysis to identify the protein cargo associated with Tub3A in the muscle. This approach could reveal potential associations with *Wnt* signaling components, providing mechanistic insights into the observed phenotype following colchicine treatment.

Minor Comments:

1. The results section, written without subheadings, is difficult to follow. I suggest breaking it into multiple subsections to improve readability and comprehension.
2. Figure 5B: The posterior-facing wound site graph has very few control samples, resulting in a high standard deviation and contributing to the significant difference observed compared to the 200 $\mu\text{g}/\text{ml}$ colchicine-treated sample. Including more control samples would strengthen the statistical rigor of this analysis.
3. Performing β -catenin FISH at colchicine-treated cut sites could provide valuable insights into the localization of the transcript in response to microtubule perturbation. This could help determine whether microtubule disruption affects β -catenin distribution, influencing gene expression and regeneration patterning.
4. In lines 356-365, all the figures referenced belong to Figure 6, not Figure 5.
5. In Figure 5B, colchicine treatment at both 125 $\mu\text{g}/\text{ml}$ and 200 $\mu\text{g}/\text{ml}$ appears to alter the spatial expression of *Wnt-1*. Could these alterations in spatial expression of *Wnt-1* explain the observed phenotypes? Specifically, the expression of *Wnt-1* in animals treated with 125 $\mu\text{g}/\text{ml}$ colchicine differs significantly from both control and 200 $\mu\text{g}/\text{ml}$ colchicine-treated animals. Could

this difference in spatial expression of Wnt-1 account for the increased Notum expression seen in the 125 µg/ml colchicine-treated condition?

First revision

Author response to reviewers' comments

Comments from the Reviewers:

Reviewer 1: SUMMARY OF THE ADVANCE MADE IN THIS PAPER AND ITS POTENTIAL SIGNIFICANCE TO THE FIELD

Anderson et al. studied the formation of a microtubule network within planarian muscle and its role in regulating injury-induced genes crucial for regeneration patterning. The specific staining of microtubules in body-wall muscles, which can be modulated by drugs targeting microtubules, is particularly noteworthy. This study used this valuable tool to elucidate how the asymmetry of body polarity is achieved through the longitudinal muscles. The findings also provide insights into the asymmetric expression of notum after injury in planarians. I believe it would be significant to show how the longitudinal muscles align across the entire animal to understand the regulation of notum expression across the scale of the entire animal, given the relatively short length of the longitudinal muscle fibers. A more in-depth focus on this aspect in the discussion would facilitate understanding. My major and minor comments are as follows.

SUGGESTIONS TO AUTHORS

1. The most significant conclusion I seek is an understanding of how microtubules within muscle fibers regulate cell polarity and the asymmetric distribution of wnt components. The authors highlight that longitudinal muscle fibers are ~ 200 µm in length, presenting a challenge in understanding the regulation of wnt components throughout the entire planarian body. Alternatively, there must be cell-cell connections to facilitate alignment through the body. I would appreciate visual evidence of the longitudinal muscle fibers to determine whether a single muscle fiber extends from the head to tail, or from the anterior wound to the posterior wound.

We appreciate this idea and agree it will be important for the field to ultimately resolve. However, we haven't yet been able to answer this particular question from the staining and imaging methods that we currently have in place. The 200 micron length value comes from other studies that looked at dissociated muscle cells, yet animals attain lengths that are much longer, suggesting that probably individual muscle cells link together in some way to form the musculature. However, in the many different whole-mount stains of planarian muscle observed across different studies including our own, it is generally quite difficult to ascertain where one cell's muscle fiber begins and another ends. While we can detect expression of TUB3A protein in longitudinal muscle fibers, we find that it is difficult to track them over long distances. We think this could reflect differences in the persistence length of muscle microtubules that do not necessarily extend across the whole fiber from each cell, or simply a difficulty in tracking because of the high density of fibers, or perhaps other reasons we have not considered. We now include movies of z-stacks (from both the Carnoy's and NAFA fix, at higher and lower magnification) to give a sense of the kind of data and challenges associated with the task of tracking fibers and cell boundaries in tissue. Ultimately we believe the answer to the question of how muscle fibers connect together will likely require other means for either specifically detecting the surface of these cells or perhaps junctional components that could be enriched at sites of muscle contact. We believe our study makes an important first step to understanding a relationship between muscle fibers, microtubules, and the Wnt polarity system driving head/tail regeneration decisions.

2. I believe that Figure 1 and 2 could be merged into a single figure. It would allow the text to be more effective in the functional discovery, as both figures are intended to demonstrate that the antibody targets the microtubules within the muscle cells. Similarly, Figure 4 and 5 would also be better to be combined into one figure.

We took this advice and merged the figures as suggested and agree it makes for a better presentation of the work. Old Figure 1 and 2 and now new Figure 1, and old Figure 4 and 5 show the experiments to find that microtubules are required for notum expression and notum polarity, as well as involvement of wnt1 and beta-catenin in this process. Our extended timeseries of wnt1 revealed an unexpected negative regulation on posterior pole wnt1 so we made the effects of colchicine on wnt1 a separate figure (new Figure 5).

3. In Figure 7, the results indicate that wnt1 RNAi combined with Colchicine treatment increased the penetrance of posterior head regeneration. However, Figure 6, demonstrates that wnt1 RNAi diminishes the posterior expression of notum when treated with 125 µg/mL colchicine. How can these results be interpreted and integrated into the pathway model? At this concentration of colchicine, reduced activin-2 and sustained wnt1 may lead to the expression of notum posteriorly. However, wnt1 RNAi leads to a reduction in posteriorly expressed notum, complicating the understanding of how these results align with the proposed pathway.

Wnt1 and notum are both functionally important for the head-versus-tail decision but in complex ways. Notum is a feedback inhibitor of Wnt signaling in general in most systems and also in planarians, and prior work has shown that injury-induced notum expression is eliminated after beta-catenin RNAi (Petersen and Reddien, 2011) and is reduced after wnt1 RNAi (Gittin and Petersen, 2022). However, wnt1 or beta-catenin RNAi animals still regenerate heads at normal anterior-facing wounds, and also at posterior-facing wounds. Therefore, notum itself can be “replaced” by conditions that mimic its role to reduce Wnt signaling. We agree with the reviewer’s interpretation here that the enhancement we detect between colchicine and wnt1 RNAi in leading to more posterior head regeneration (Figure 6 in the revised version) cannot be explained only as a result of ectopic notum because Figure 4 shows these conditions do not involve ectopic notum. However, this overall result is similar to other treatments previously reported to enhance the phenotypic effects of wnt1 silencing on head/tail identity and also decrease injury-induced notum expression, for example co-inhibition of wnt1 with wntP-2 (taken together from the results of Petersen and Reddien 2009, Gittin and Petersen 2022).

In this revised version, we now monitor wnt1 expression in colchicine-treated animals for several days during posterior regeneration and find that wnt1 expression at the posterior pole is still intact despite decreases to wnt1 at the injury-induced phase at 12 hours (Fig 5A,B). Animals treated with colchicine homeostatically for several days also actually displayed a slight increase in the wnt1+ domain (Fig 5C). Together these changes to wnt1 provide another candidate mechanism to explain the enhancement observed between microtubule inhibition and wnt1 RNAi (Fig 6). Our narrow interpretation here is that microtubules in some way act with wnt1 in order to promote tail identity, which is demonstrated by Figure 6.

4. It is intriguing to consider how the morphology at the flank incision wound site was preserved during the fixation and staining procedures. Technically, this understanding is critical for accurately interpreting the results.

We used Carnoy’s fixative under standard procedures but tried to handle the animals gently through the process. We now update this information in the Methods Section.

5. The transition from the suppression of notum expression to the ectopic expression at the posterior wound is “surprising”. It would be beneficial to provide additional data to support this observation, as all the RNAseq data analysis are from the 200 µg/mL Colchicine treatment. If the recovery of microtubules during regeneration explains the normal regeneration outcome, it is essential to specify the days post injury when the microtubule recovered to normal levels. For instance, at 5 dpa the authors showed a delay in regeneration after treatment of colchicine for 24 hours and subsequent regenerate in control media. Have the microtubules recovered at this time point, and what is the expression of notum at the posterior pole?

We too were very surprised by this result because we initially predicted that microtubule inhibition would lead only to nonspecific effects. The observation that low doses of nocodazole or colchicine lead to excess notum at posterior-facing wounds was obtained through a candidate approach and was actually the first observation made in the project (hence the extensive amount of analysis that

went into understanding it). We now provide additional support for this result, which we have observed in 94 animals probed across 4 experiments shown in Figure 4A, 2 experiments shown in Figure 4B, and an additional 4 separate experiments in the supplemental figures.

We really appreciate the suggestion to test the role of microtubule recovery after the dosing. We had initially chose this experimental design based on the McWinnie experiments which used a similar strategy to inhibit microtubules for a few days prior to injury and then allowed them to recover in colchicine-free media. McWinnie did this to avoid lethality observed from longer colchicine treatments so that any specific effects on regeneration could be assessed, and we find that *S. mediterranea* is under the same overall constraints. We agree though that this design leads to the formal possibility that effects on *notum* or *wnt1* expression could arise from the recovery following release from microtubule inhibition rather than microtubule inhibition itself. To test this possibility, we tested the impact of removing the washout from the procedure (ie, incubating for 24 hours prior to amputation then placing in fresh colchicine after surgery until fixation at 18 hours). We observed that this treatment still produces ectopic notum expression (Figure S13). Therefore, we believe this particular effect is due to microtubule inhibition rather than the release from inhibition. For completeness we also verify that higher concentrations of colchicine causing reduced notum and *wnt1* also still occur in animals treated continuously (Fig S13).

In related experiments, we previously tried to ascertain how long recovery after microtubule inhibition drug treatment would take through staining with the Tub3A antibody. At the doses we tried, the microtubule system seemed still roughly as disrupted at 5 days recovery as at 1 day (not shown). However, this experiment seems imperfect because it may be difficult to detect very many different degrees of microtubule recovery when the initial treatment at these doses does not fully eliminate existing muscle microtubules. Also we have no way to measure how quickly washout removes colchicine that may reside latently in or throughout tissue. In principle, another way to address the issue of effects on regeneration could be to continuously dose animals during 5 or more days of regeneration. However, like McWinnie studying another species of planarian, we found that dosing at 150ug/ml and 200ug/ml for this long killed all the *S. mediterranea* animals.

We also now show the kinetics of notum and *wnt1* expression in colchicine-treated animals. We find that anterior pole formation is quite delayed, with only a few animals having a notum+ anterior pole at day5 (Fig 4B). This is consistent with observing slower head regeneration overall and also the experiments showing that anterior *sfrp1* expression did not emerge until a later time by day 14. We think it is likely this effect could be attributed to the mitotic arrest effects of colchicine, which seem to have an anterior bias (Fig S2), because it is known that anterior pole formation requires proliferation. As colchicine-treated animals regenerated posteriorly, they never expressed pole-like *notum* or *sfrp-1*, and never formed posterior heads. By contrast, colchicine-treated animals had prominent expression of *wnt1* at the posterior dorsal midline. This expression was evident from the early stages of regeneration (18 hours) and sustained from 48-96 hours, and actually involved anterior expansion of the *wnt1*+ posterior domain (Fig 5C-D). We verified these effects also occurred after 3 days of homeostatic colchicine treatment as well, although without a detectable impact to other posterior patterning genes. We suggest that the ability of colchicine-treated animals to activate *wnt1* at the posterior midline could enable them to regenerate posteriorly even under conditions that strongly reduce the early expression phase of wound-induced *wnt1*. This model then offers a candidate explanation for why co-inhibition of microtubules and *wnt1* could increase the phenotype of posterior head formation, if wound-induced and posterior pole *wnt1* expression can jointly promote tail fate.

6. In lines 359 and 362, the referenced figures should be Figure 6A and 6B.

We fixed this error.

7. Figure 1D shows the muscle cell body surrounded by TUB3A proteins. Conversely, I did not observe similar patterns in Figure 1E. Is this difference due to the two staining methods or is it merely a result of not displaying the images in a similar manner.?

The colocalization of TUB3A surrounding nuclei in this way was indeed observed only rarely. We have now included z-stack movies to give readers a more complete picture of the nature of labeling (including both Carnoy's and NAFA-fixed animals). Movie 3 shows the immunostaining of Tub3A plus

FISH of collagen (NAFA fix) and some green TUB3A+ signal can be detected close in the vicinity of the muscle cell bodies. However, we think that the fact that the fiber plexus is in a different D/V z-plane than the muscle cell bodies and also the sheer number of fibers present make it challenging to detect these associations in general. We updated the model to not include the microtubules surrounding each cell body and updated the explanations for the observations in the text.

8. The image resolution in Figure 2A is inferior to that in Figure 1. Please address this issue. How many animals were used as controls in Figure 2B? Additionally, interpreting the black upper right corner in the right panel of Figure 2B for 6G10 and TUB3A co-staining is challenging. Were there anything issues following the laser bleach?

The samples in the original Figure 2A (now Fig 1F) were stained with tyramide and imaged at lower magnification (20x) rather than as in Fig 1B which were imaged by direct labeling and imaging at 40x/1.25NA glycerol objective with 6x confocal zoom. We found that direct labeling produced the sharpest results, probably because of the extremely small size of the microtubules (note they are substantially thinner than 6G10+ stains of muscle fibers), but the downside was that samples could only be imaged for about 1-2 weeks with the best quality for staining. Tyramide labeling was less resolved but the samples could be stored and imaged over a much longer time before loss of label, which was useful in some cases in order to have enough time to image many replicate samples for making robust conclusions, for example in the timeseries after incision (Fig 2) or amputation (Fig S3). We now clarify this difference in imaging/labeling in the methods and also note in Figure 1-2 legends how TUB3A was detected in each case. We also reimaged some of the panels in old Figure 2A (new Figure 1F) to improve clarity and now include supplemental movies showing z-stacks of the imaging.

We have also now added numbers of control animals in the original Figure 2B (now Fig 1G).

The black area at upper right corner of original Figure 2B (now Fig 1G) was at the edge of the animal and so the tissue curved in that region, such that the z-stack selection did not capture this area, so we re-cropped the images to display the range suitably detected at the same D/V position (Fig 1G). We have not noticed any particular issues with bleaching after detection of microtubules versus other tissues and stains in planarians.

9. What is the change in expression of activin-2 after colchicine treatment at 125µg/mL? The data presented in figureS10 are for 200µg/mL, not 125µg/mL.

Through three different revision experiments we set out to conduct at 125 and 200 µg/mL colchicine, we found that the observation of increased activin-2 between 0 and 18 hours in the wound sites of control animals was not robust. Without this change from 0h as a baseline change following wounding, it was difficult to detect a decrease in activin-2 expression after colchicine treatment by tyramide FISH. Therefore, we removed this supplemental figure and the associated interpretations from the text.

10. Determining whether notum is the only gene increased in the PFW may be beyond the scope of this study. However, it is quite interesting to conduct RNAseq on the PFW tissues to confirm the decrease in *wnt1* and the increase in *notum*.

We agree that the datasets presented here will give a wealth of information for future studies. *notum* is the only gene to be activated at normal anterior-facing wounds (Wurtzel et al 2015) so it is possible that the suggested experiment might only confirm the unique nature of *notum* asymmetry.

11. In the Methods Section, additional details are required. please specify whether the worms are maintained in a circulation system. The dsRNA concentration needs to be specified.

The catalog number of each reagent should be provided.

Additionally, clarify what is PBSTi represents.

We now clarify: (1) animals were maintained in static culture. (2) dsRNA was fed at a concentration of 300 ng/ul. (3) We also now add product numbers to the methods section. (4) we clarified that PBSTi stands for PBSTx (0.3% TritonX-100) plus imidazole (10 mM).

12. Due to the unavailability of the Tables for verification, I have not confirmed this part of data.

We apologize that this must not have been available on the first submission but should now be accessible.

Minor points:

1. There are a couple of corrections in the References:

Please ensure the correct use of the Latin alphabet in authors' name. For instance, "Sanchez" should be corrected to "Sánchez", and "SALO" should be corrected to "SALÓ". Please make these adjustments throughout the references.

Line 1125 and 1128, the authors cited the same paper as two references. Please ensure the authors' names are formatted consistently. For example, ALVARADO, A.S., or SANCHEZ ALVARADO, A.

Thanks for catching these errors which we now believe are fixed.

2. Understanding whether the anti-TUB3A staining exhibits distinct staining signals under the two different fixation protocols is beneficial.

The antibody did not work under formaldehyde fixation but we have not noticed qualitatively different stainings between the Carnoys method and the newly introduced NAFA fixation - both detect expression within muscle fibers. Please note that Figure 1B was imaged to a much higher magnification than the surrounding panels in order to highlight the unique staining pattern of TUB3A filaments surrounding the fibers. Also note that the NAFA fix and imaging we performed used tyramide labeling which as described above increases signal and is useful for long-term storage of labeled specimens for repeated reimaging, but also decreased some of the sharpness of signal we could observe by labeling as in 1B with fluorophore-conjugated secondary antibodies.

3. The scale bars of Figure 1E are missing.

We added these in the revised version.

4. Is Nocodazole media also prepared freshly, similar to the colchicine media?

We prepared nocodazole solutions freshly before use and have updated this in the Methods.

5. When authors claim the "muscle cell body could be a site of tubulin monomer accumulation", please clarify that colchicine is to inhibit the tubulin heterodimer from polymerization. It should be noted that this is a site of tubulin heterodimer accumulation.

Thank you for identifying this error. We corrected this statement to read that "muscle cell body could be a site of unpolymerized tubulin heterodimer accumulation"

6. The thickness of the scale bars should be consistently maintained in Figures.

We have now tried to fix this issue throughout and have uniformity within and across each figure.

7. Please adjust the font of μg in figures.

We fixed the font issue

8. Muscle fiber regrowth across wound sites has recently been reported in Lu et al., eLife, 2024, which can be considered to be included in the citation.

We appreciate drawing our attention to this nicely complementary study and now have cited it.

9. TUB3A or Tub3A should be consistent in Figure 3 and throughout the manuscript.

We fixed this to have the consistency: TUB3A to refer to the protein, *tub3A* to refer to the gene or mRNA.

10. H3P cells should be H3P+ cells. mm² should be mm2.

We fixed these typos in the text and figures

11. A 2-tailed unpaired t-test should be referred to as a two-tailed unpaired t test. Is this Student's t test? If so, please specify accordingly.

In the original submission they were all unpaired two-tailed t-tests. However, based on another reviewer's comment, we redid all of the statistical analyses because in most cases there were multiple comparisons taking place, necessitating ANOVA or non-parametric tests (Dunnett's and Dunn's tests as necessary) which are now indicated in the legends plus Methods. All of these were two-tailed tests.

12. In the legend of Figure S5, there is a redundant description of runt-1 expression. Please remove one instance.

We fixed this.

13. On line 614, a citation is labeled as [29], which is not consistent with other citations.

We fixed this citation issue.

Reviewer 2: Review Manuscript: dev.204669, Anderson & Petersen

During regeneration, cells must undergo dramatic and dynamic changes in cell shape and identity. Though the cytoskeleton has long been known to regulate diverse cellular processes, including cell migration and cell shape, it has not been well characterized in whole-animal regeneration processes. In this paper, Anderson and Petersen have investigated specific properties of the microtubule cytoskeleton in planarian flatworms, exploring the role of microtubules in muscle cells in regeneration. Importantly, muscle cells in planarians have typical contractile roles in animal movement and wound healing, but they also serve as critical signaling sources of cues necessary to repattern animal tissue after amputation. This paper fills a much needed gap in our understanding of the cellular biology of muscle cells in planarians and in the role of the cytoskeleton in driving specific aspects of regenerative biology. Strengths of the manuscript include: 1) its innovative focus on cell type-specific cytoskeletal proteins to shed light on microtubule biology in planarian muscles; 2) a careful return to and replication of prior experiments perturbing microtubules using chemical inhibitors; and 3) rigorously quantified data (including a fully validated new antibody). One weakness is the lack of clear mechanism connecting microtubules to polarity signaling. This weakness is mitigated to a large extent for overall enthusiasm about this exciting project. This paper is a strong fit for Development, though a number of concerns could be addressed prior to publication.

Major:

How colchicine affects muscle cell biology and animal-level regeneration was a bit underdeveloped. More information on the overall animal response to colchicine and how cells are impacted could help shed light on the results with Notum and polarity.

1. Are muscles able to contract sufficiently to allow timely wound closure after lateral cut or amputation injuries? Some figures like S3 show features that might be interpreted as slow wound closure.

We now present an analysis of wound closure after colchicine RNAi by imaging live animals after amputation (Fig S7). Wound sites initially allow material from the inside of the animal to escape and have a jagged appearance but as the wound contracts it takes on a smooth appearance and outflow of material stops. Colchicine-treated animals healed their wounds on a similar timescale as control animals. We're not aware of an existing assay to determine specifically the contraction of muscle at a wound site, but because healing occurred we infer it was likely normal.

2. Is wound response (in terms of gene expression) changed temporally after colchicine treatment. Some of the results in Fig. 4 might be consistent with a delayed or prolonged wound response, which would likely change the reader's interpretation of these results.

We now present plots from the RNAseq analysis showing that indeed a complex set of changes happens to wound-induced genes after colchicine treatment (Fig 3B, Fig S4C). Some genes had little to no activation (*wnt1*, *notum*, *runt-1*, *h2b*), while many others had a relatively normal behavior (eg., *inhibin* and *wntless* which we verify by FISH in Fig 3C). The fact that *inhibin* and *wntless* are activated in muscle also indicates that muscle from colchicine-treated animals is capable of undergoing wound-induced gene expression with normal kinetics. Other genes underwent activation at seemingly normal kinetics but began with a higher baseline of expression (eg., *egr2*). Based on these results, we think it is unlikely that the effects of colchicine on *wnt1* and *notum* are likely to be explained only by a wholesale delay in all injury-induced gene expression activation.

3. Are small blastemas seen after colchicine treatment and regeneration? For pieces with a short A/P length, do any blastema defects (e.g. split blastema) or reversed polarity phenotypes occur as was seen in McWhinnie, 1955? Do these require continuous colchicine administration or is the 24 hour period enough?

We have tried a variety of dosing paradigms seeking outcomes similar to what McWhinnie observed (using a different species of planarian) but never saw posterior head regeneration in *S. mediterranea*. We suggest the difference in outcomes is likely due to species-specific differences in the polarity mechanisms. As a candidate explanation for why colchicine alone did not cause posterior head regeneration, using the timecourse of *wnt1* and *notum* expression (Fig 4A-B, Fig 5A-B), we detect that although colchicine-treated animals have reduced early *wnt1* wound-induced expression, the late expression phase is still intact (Fig 5B) and actually expanded. *Wnt1* pole formation and microtubule-dependent wound-induced *wnt1* expression could act in parallel pathways such that colchicine treatment enhances the effect of *wnt1* RNAi to reverse polarity in regeneration (Fig 6A, C).

We now also present an experiment showing that indeed continuous exposure to colchicine (24 hours prior to injury and also for 18 hours after injury) causes similar effects on *wnt1* and *notum* in animals fixed at 18 hours post amputation (Fig S13). Like McWhinnie studying another species of planarian, we find that longer-term exposure of *S. mediterranea* to colchicine kills animals after about 4-5 days, which prevented an assessment of whether head/tail regeneration polarity could be flipped at high concentrations of colchicine or through continuous exposure.

4. Is posterior signaling (e.g. wnts) normal at 12dR after colchicine treatment? The tail looks a bit underdeveloped in the live animal image.

We now show that *wnt1* posterior pole expression still occurs in colchicine-treated regenerating animals and is actually expanded anteriorly (Fig 5B). Further tests found that homeostatic inhibition of colchicine for 3 days expanded the *wnt1* domain in uninjured animals though without much effect on other posterior Wnt genes (Fig 5C, Fig S12A). We think it is likely that the regeneration delay phenotypes could in part be due to recovering from mitotic arrest, but that the phenotypes of enhanced ectopic head regeneration (by coinhibition of Wnts and microtubules) are unlikely to be due to an effect to decrease stem cell activity overall.

5. What is happening to muscle cell shape and connectivity after colchicine treatment in the presence of injury? The authors note decreased 6G10 staining. Are other antibodies available to stain MHC (e.g. TMUS-13 used by F. Cebrià, 2000) to take a look at actomyosin? Can cells be dissociated to see if muscle cells maintain their shape or length (as per Witchley, 2014)? These experiments would help to understand exactly the impact of microtubules on muscle cells.

We unfortunately do not have access to the TMUS-13 antibody at this time and worried it would be challenging to ship reagents overseas right now (from Spain where this antibody originated) because of issues with customs in the US. Instead, we tried examining actin with other immuno/affinity reagents that can be effective across many different organisms: JLA20 anti-actin antibody, AAN02-S anti-actin antibody, A4.840 anti-MHC antibody, MF14 anti-MHC antibody, and phalloidin. However, we were not able to detect signal under conditions (Carnoy's or NAFA fixative) that also allow for detection of TUB3A using the antibody we generated. We suspect this is because the detection of TUB3A with our antibody likely requires some denaturing conditions that are not compatible with many existing reagents for MHC detection. We have not yet tried cell dissociations because we were concerned that formaldehyde fixing them would not be compatible with the TUB3A antibody, and it was unclear to us whether Carnoy's fix or NAFA would work well on these kinds of dissociated cells. However, we believe the evidence we currently provide definitively establishes that Tub3A is expressed in muscle.

Minor:

1. (Optional) I found myself wondering more about the genes differentially expressed at the 0h time point. This could be addressed in a supplemental figure for others in the field who are interested. In which cells were these genes expressed (predominantly muscle or a mix? What categories (GO terms, for example) are overrepresented? I think this is worth including so that others can build on this interesting work!

We apologize for the error as there seemed to be a supplemental file reporting all of the RNAseq results that was not accessible during the first submission but which we now hope is available for further analysis. As suggested, we have plotted the expression values over time for a variety of known injury-induced genes and this highlights the complex set of responses to colchicine. We find evidence that colchicine strongly prevents expression of some wound-induced genes (notum, wnt1, runt1, h2b), but only moderately or minimally affect others (inhibin, wntless, equinox, follistatin, nlg1). There is a broader category of factors such as egr2 that are elevated at the onset of regeneration (0h) but still undergo induction. The colchicine-dependent responses span several cell types but there is not a single type of wound-induced gene category (early/late, muscle vs epidermis, etc) that is fully lost after colchicine treatment. A very large number of genes are differentially regulated at the 0-hour timepoint and we hope that these supplemental files will be useful to the field for others to mine for unraveling the diverse effects of colchicine treatment on whole planarians.

2. There are some statistical errors:

a. t-tests are not appropriate for experiments with 3 samples (e.g. Fig. 5 and similar). Instead, one-way ANOVA with a post hoc pairwise test would be appropriate for this type of experiment.

We appreciate alerting us to this issues. We now use tests designed for comparison of multiple samples for all statistical analysis where they are needed. In one case we were able to use a one-way ANOVA as suggested (Fig S2). However, we found that other experimental datasets did not meet the criteria of ANOVA for normal distribution (Shapiro's test) and/or equal variance (Levene's test), so instead in those cases we used a nonparametric test (Kruskal-Wallis test) followed by post-hoc Dunnett's test the case of comparing multiple samples to a common control condition.

b. t-tests are not appropriate for experiments with 2 variables (e.g. Fig. 6 and similar). Instead, a two-way ANOVA would be more appropriate.

We appreciate this suggestion and redid the statistical analysis as suggested. In cases of samples with two variable not meeting the criteria for normally distributed data (by Shapiro's test) and equal variance (Levene's test) we used a nonparametric test aimed at making multiple comparisons across many groups (Kruskal Wallis test followed by Dunn's test).

3. Occasionally, the text indicates that cytoskeletal filaments are being observed rather than cells.

a. Line 74, "Planarian muscle cells possess actinomyosin contractile fibers that project..." It is the muscle fibers consisting of end-to-end cells rather than the contractile fibers that project across the body.

We appreciate this comment but are uncertain about what is the unclarity here. As far as we know there is not yet any literature on the subject of whether planarian muscle cells make end-to-end contacts, though we agree it is the most plausible way for the L-muscle system to link together to form a seemingly continuous set of projections across the body. We may also be mistaken or in disagreement about the definition of the components of planarian muscle cell anatomy here, but felt that a reasonable definition of fiber is the contractile projection present on each muscle cell, rather than a description of the supracellular network of adjoined fibers (if this exists). We have used “contractile fiber” and “muscle fiber” interchangeably because the fiber projection from each muscle cell is capable of contraction. However, if there is a more accurate way to describe or define these terms within the field, we are happy to do so. For additional clarity in the sentence referenced above, we replaced “actinomyosin contractile fibers” with “actinomyosin-rich contractile fibers”.

b. Line 236, “TUB3A+ microtubules could be observed...” It is probably the muscle cells being observed at this magnification. (See also lines 238 and 240).

We clarified this in the text as “TUB3A+ muscle fibers”

4. Using magenta and green rather than red and green can help readers with color vision to see the beautiful staining data better.

We made this change and agree it stands out better this way.

5. There is a typo in regeneration in Fig. 8.

We fixed this typo.

Reviewer 3: The manuscript by Anderson et al., titled “Planarian Microtubules Form a Network Within the Muscle and Regulate Injury-Induced Genes Essential for Regeneration Patterning,” explores the role of microtubules in regulating the expression of wound-induced genes, specifically *wnt-1* and *notum*, which are critical for establishing anterior-posterior (A/P) polarity during regeneration. The authors identify a muscle-expressed tubulin isoform, *Tub3A*, but its knockdown does not result in a discernible phenotype. Nevertheless, the authors attribute the microtubule function to colchicine treatment, which depolymerizes microtubules, leading to the downregulation of muscle-expressed, wound-induced genes and resulting in defective A/P polarity. Although the authors present several important observations, many of their conclusions are based on correlational evidence and remain highly speculative. The manuscript lacks a clear mechanistic explanation for how microtubules regulate wound-induced gene expression.

We appreciate the helpful suggestions. We have addressed all of the major comments through incorporation of new experimental evidence supporting the model that microtubules participate in the fating of regeneration blastemas by controlling injury-induced expression of *wnt1* and *notum*. We agree that understanding how microtubules accomplish these tasks will be a very important direction. Though given the very large number of tubulins and associated regulatory proteins in planarians (~60 alpha tubulins, ~60 kinesins and a host of other associated regulatory factors, etc), and also the difficulty that no transgenic approaches are yet available in planarians to readily detect the localization of proteins in that animal, we believe this will be a formidable challenge beyond the scope of what one study can accomplish. However, we note that despite recent work showing the importance of muscle as a key information source in regeneration through responding to injury and signaling missing tissue, still vanishingly little is known about its basic cell biology and also the signaling processes directing its activities. *Wnt1* and *notum* are crucial regulatory control points, but there is still a lack of information about the molecular mechanisms that activate and regulate the expression of these genes. We believe in this context, our work will be important for the field as a foundational study for revealing the nature of muscle microtubules as oriented along fibers, as well as the functional experiments we conduct to show participation in the Wnt and related pathways for control of regeneration.

Major Comments:

1. The authors use different concentrations of colchicine to perturb microtubule polymerization. Since microtubules are essential for muscle function and integrity, and are also involved in the transport of proteins and RNA, it is not surprising that perturbing such a crucial molecule could have long-lasting effects on muscle integrity. The phenotypes observed could, therefore, be due to disrupted muscle integrity rather than a direct role of microtubules in regulating Wnt-1 and Notum expression.

We now provide independent lines of evidence that microtubule disruption affects *wnt1* and *notum* expression rather than causing a loss of the cell type or broad dysfunction of the muscle cells expressing these factors. First, colchicine-treated animals still have collagen⁺ cells and myoD⁺ cells at normal abundance (Fig S9). In addition, colchicine treatment did not eliminate expression of other wound-induced genes also expressed from muscle (eg., *wntless*, *inhibin-1*) and we provide both FISH and RNAseq to demonstrate this. Also, while the formation of new 6G10⁺ muscle fibers through wound repair was impaired by colchicine, the broad pattern of 6G10 staining was relatively normal throughout the animal, suggesting that these doses of microtubule inhibition did not fully eliminate muscle integrity (Fig 2). An intriguing possible explanation of the results, that we are not able to resolve more fully without new tools and approaches, is that some, but not all, injury-induced genes expressed in muscle require fiber regrowth for expression, and our results represent a first step for the field to further understand the complexity of the wound response in planarians. Finally, the microtubule-inhibited animals did show signs of slower regeneration (for example, delays in *notum* / *sfrp1*⁺ anterior pole formation) (Fig 5B, Fig S10), but we found that this treatment could interact in a quite specific way with *wnt1* inhibition to enhance the regeneration of ectopic heads (Fig 6). We note that the phenotype of posterior head regeneration is highly specific - in an RNAi screen of over 1000 genes none caused this effect (Reddien 2005)- and there are only a handful of treatments described in the literature that result in this defect or modify it. While we certainly agree that microtubule inhibition has many diverse effects on the animal, our results collectively argue for microtubules exerting some regulatory action related to *wnt1* and *notum* expression, and the control of blastema specificity.

2. The authors show that 200 ug/ml colchicine treatment causes more global defects such as cell cycle defects etc, whereas the 125 ug/ml colchicine concentration seems to affect muscle more specifically. However, it can be argued that 125 ug/ml colchicine might perturb specific muscle types more than others. To address this, the authors should examine the expression of other muscle-specific markers, such as *myoD* and *nkx1-1*, to assess overall muscle integrity. Furthermore, the authors should also test the expression of additional polarity genes, such as BMP and *Slit*, which mark dorsal/ventral (D/V) and midline regions. Perturbation of microtubules might also affect these genes, in addition to A/P markers like Wnt-1 and Notum.

We examined *collagen* (pan-muscle) and *myoD* (longitudinal muscle), and found that these types of muscle were not reduced by colchicine treatment (Fig S9). Therefore, the lack of *wnt1* or *notum* expression is not likely to be a result of the absence of responding cells. We also note that some other genes induced to be expressed in muscle (*wntless*, *inhibin*) are not perturbed by colchicine treatment (Fig 3, S4), indicating that microtubule inhibition still enables muscle to respond to wounds. We also tested the patterning gene *slit* and found *slit* expression seemed normal in animals at the time microtubules are required for injury-induced *wnt1* and *notum*, arguing it is unlikely that microtubules act through *slit* to control *wnt1/notum* (Fig S12). Although our analysis shows that the *wnt1* and *notum* expression control mechanisms involve microtubules in specific ways, both the 125 and 200 ug/ml colchicine doses probably cause many effects in the organism that are likely to be unrelated to blastema fating but our analysis indicates there are likely specific regulatory pathways microtubules act upon to regulate *wnt1* and *notum*.

3. I am surprised that transcriptome sequencing was not conducted following treatment with 125 ug/ml colchicine. I recommend performing transcriptome sequencing for both 125 ug/ml and 200 ug/ml colchicine treatments and comparing the results. This would allow identification of specific polarity markers and muscle types that are more dramatically affected by each treatment.

We appreciate this suggestion. The flow of discovery actually started with observing the ectopic *notum* expression after lower doses of nocodazole/colchicine, then subsequently we found the high doses caused complete loss of *wnt1* and *notum*, then we performed RNAseq in order to better understand the specificity of this second phenomena to discover that *wnt1* and *notum* are

surprisingly among the most differentially regulated genes at that treatment. The reason we did not undertake RNAseq to examine the excess notum expression phenotype is partly technical, partly biological. First, while the effect of 125 ug/ml colchicine to increase notum expression at posterior-facing wounds is quite reproducible (note we now support Figure 4A with a combination of 6 independent experiments on n=59 animals), the maximum penetrance of defect we typically observe for detecting this effect in any one experiment ranges from ~25-40% of animals. Note too that while we can find some rare animals that express up to 50 notum+ cells at posterior-facing wounds treated with 125 ug/ml, the average is still only ~10 notum+ cells. We worried that in a bulk RNAseq experiment requiring many samples and replicates that the average increase to notum at these wounds would likely be low, so in the effort to conserve resources during a time of upheaval to science funding (XA's NIH F31 fellowship was terminated by the government, and also all federal funding to Northwestern including CP's funding has been frozen since March 2025), it was an experiment we did not attempt and instead prioritized addressing other concerns. Second, based on other prior work (Wurtzel 2015), *notum* is the only gene with asymmetric wound-induced expression in planarians, so we viewed it as unlikely that such a dataset would do more than merely confirm overactivation of this one gene. Instead, we have tried to consider all possible mechanisms involving notum regulation as reported in the literature. Based on responding to another reviewer comment, we now performed more extended timeseries to monitor the effects of colchicine on *wnt1* (Fig 5A-B). First, this analysis found that *wnt1*, known to facilitate notum, is delayed in its expression onset at posterior-facing wounds at the 125 dose, providing a candidate explanation for why notum is upregulated under these conditions. We believe then, the major mechanistic questions for the future would actually be how microtubules promote *wnt1* expression, which could be further evaluated with the RNAseq dataset we already provide.

4. Since the authors have raised an antibody against the muscle-specific tubulin, Tub3A, it would be valuable to perform pull-down assays with this antibody followed by proteomic analysis to identify the protein cargo associated with Tub3A in the muscle. This approach could reveal potential associations with Wnt signaling components, providing mechanistic insights into the observed phenotype following colchicine treatment.

Certainly it will be very interesting to identify how microtubules link to the signaling pathways involved in regeneration. However, based on the biology and biochemistry, we think this particular approach is not likely to be fruitful. Of foremost concern is that polymerized tubulin is unlikely to be soluble, which would be a prerequisite for successful immunoprecipitation. Also, the fact that this antibody only works through denaturing fixation (Carnoy's) and not a typical formaldehyde fixation might suggest it is unlikely to bind to the folded protein. In addition, even if immunoprecipitation were to be successful there are many tubulin-binding proteins, and we believe that the ones relevant for function modification of Wnt signaling would be difficult to disentangle from those engaged in other biology.

Minor Comments:

1. The results section, written without subheadings, is difficult to follow. I suggest breaking it into multiple subsections to improve readability and comprehension.

We added subheadings and agree this improved the flow and organization.

2. Figure 5B: The posterior-facing wound site graph has very few control samples, resulting in a high standard deviation and contributing to the significant difference observed compared to the 200 µg/ml colchicine-treated sample. Including more control samples would strengthen the statistical rigor of this analysis.

We repeated the experiments and added more control and experimental samples (now all samples have n>15 and across three experiments) to the section so that the effects are more robustly supported.

3. Performing β -catenin FISH at colchicine-treated cut sites could provide valuable insights into the localization of the transcript in response to microtubule perturbation. This could help determine whether microtubule disruption affects β -catenin distribution, influencing gene expression and regeneration patterning.

We performed beta-catenin FISH with or without colchicine, but the expression was weak/broad in both cases and it did not seem re-localized. However, an important area of future work will be to determine how microtubules in muscle might influence Wnt signaling.

4. In lines 356-365, all the figures referenced belong to Figure 6, not Figure 5.

We fixed this

5. In Figure 5B, colchicine treatment at both 125 µg/ml and 200 µg/ml appears to alter the spatial expression of Wnt-1. Could these alterations in spatial expression of Wnt-1 explain the observed phenotypes? Specifically, the expression of Wnt-1 in animals treated with 125 µg/ml colchicine differs significantly from both control and 200 µg/ml colchicine-treated animals. Could this difference in spatial expression of Wnt-1 account for the increased Notum expression seen in the 125 µg/ml colchicine-treated condition?

We now present an extended timecourse of wnt1 and notum expression in regeneration to address these issues. In the case of wnt1, which ordinarily peaks at 12 hours and is declining by 18 hours, the high dose (200 ug/ml) prevented wound-induced expression at both 12 and 18 hours. By contrast, the lower dose (125 ug/ml) indeed delayed wnt1 expression from onsetting until after 12 hours such that by 18 hours it is present mainly at posterior-facing wounds. Because notum expression requires wnt1 and beta-catenin, we believe these results provide an explanation for why notum expression elevates at posterior-facing wounds only at the 125 ug/ml dose and not the 200 ug/ml dose. We also now provide more characterization of the other noted phenotype on wnt1 expression after colchicine, involving expression at the posterior midline (only on the dorsal side of the animal). This posterior-pole expression of notum normally occurs around 48-72 hours in control animals but appeared earlier and actually expanded in colchicine-treated regenerating animals. In uninjured animals treated with colchicine, this same effect occurs within 3 days. Other studies previously have found several factors whose inhibition increases the posterior wnt1 domain similarly (mob4, striatin, bmp, yki, pak1) so we suggest microtubules may act in any/all of these processes as well. We hypothesize that normal tail formation could involve joint contributions from wound-induced wnt1 as well as posterior-pole specific wnt1 and microtubules operate (oppositely) in each process, providing an explanation for the posterior head phenotype enhancement we see after knocking down wnt1 and inhibiting microtubules.

Second decision letter

MS ID#: dev.204669R1

MS TITLE: Planarian microtubules form a network within muscle and regulate injury-induced genes essential for regeneration patterning

AUTHORS: Xavier Anderson; Christian P. Petersen

ARTICLE TYPE: Research Article

Dear Chris,

I am happy to tell you that your manuscript has been accepted for publication in Development, pending our standard publication integrity checks.

Reviewer 1

SUMMARY OF THE ADVANCE MADE IN THIS PAPER AND ITS POTENTIAL SIGNIFICANCE TO THE FIELD

Anderson et al. studied the formation of a microtubule network within planarian muscle and has provided additional key results to show the regulation on injury-induced genes crucial for regeneration patterning. The antibody for staining of microtubules in body-wall muscles will

become a valuable tool to the field. This work is pioneer and will become the foundation to address the remaining questions. Overall the work has nicely addressed all of my prior comments and enhanced the rigorous of the data quality.

SUGGESTIONS TO AUTHORS

I do not have further questions.

Reviewer 2

This is an excellent revision of an already very exciting paper. I recommend acceptance.

I only have one comment to the authors, which does not need to be addressed prior to acceptance. I found it very interesting that some polarity control genes had impaired expression after colchicine treatment and I am very curious about the difference between the genes affected and those unaffected. Could it be that only a subset of muscle is affected (e.g. longitudinal)? This will be fun to look at in the future!

Reviewer 3

I have reviewed the revised version of the manuscript and the authors' rebuttal. I am satisfied with the revisions and recommend the work for publication.